# BUILDING FLOW UNIQUENESS IN ONE-STEP GENERATIVE MODELING

## ABSTRACT

Recent advances in generative modeling frameworks, such as diffusion models and flow matching, have achieved record-breaking performance. Nevertheless, these approaches involve iterative sampling procedures across many neural network passes, which severely limits their practical deployment, particularly in domains demanding real-time interaction. Although considerable effort has been devoted to accelerating sampling, achieving high-quality one-step generation remains an open challenge, motivating research into a new era of generative modeling. Motivated by this, we put forward a novel and effective framework, termed *Flow Uniqueness Models* (**FUM**). The core idea of FUM is to construct strictly one-to-one image pairs, thereby enforcing velocity uniqueness along the entire sampling path, which forms as the foundation for few-step sampling. By leveraging this modeling mechanism, FUM not only achieves remarkable one-step generative performance but also provides the flexibility to balance image quality against the number of sampling steps. Extensive experiments on three benchmark datasets comprehensively validate the superiority of our proposed FUM.

## 1 INTRODUCTION

The objective of generative modeling is to construct a path from a prior distribution to the target data distribution (Lipman et al., 2023), with sampling performed by traversing this path along a specific trajectory. Recent advances, primarily driven by Diffusion Models (Karras et al., 2022; Rombach et al., 2022; Lu et al., 2022) and Flow Matching (Liu et al., 2023a; Albergo & Vanden-Eijnden, 2023), have set new benchmarks for generating high-fidelity samples across diverse domains, including images (Ma et al., 2024; Liu et al., 2023b; Geng et al., 2025a), video (Bar-Tal et al., 2024; Kong et al., 2024), audio (Tian et al., 2025), and 3D scenes (Szymanowicz et al., 2025; Sun et al., 2025). Despite their impressive capabilities, the practical utility of such models remains constrained by a critical bottleneck—the reliance on iterative sampling, which typically requires tens or even hundreds of neural network inferences (Zhang & Chen, 2023; Zhao et al., 2023; Liu et al., 2022; Tong et al., 2025). This limitation poses significant challenges for real-world deployment, particularly in resource-constrained or latency-sensitive environments (Li et al., 2023).

In response to this dilemma, a rapidly growing body of work (Zhou et al., 2025; Wang et al., 2025b; Frans et al., 2025; Zhang et al., 2025; Song et al., 2023) has focused on developing few-step or even one-step generative models aimed at drastically reducing sampling costs. These approaches can be broadly categorized into three main groups, *i.e.*, diffusion distillation (Nguyen & Tran, 2024; Yin et al., 2024b; Salimans et al., 2024), trajectory consistency (Lu & Song, 2025; Song & Dhariwal, 2024; Wang et al., 2024a), and flow rectification (Esser et al., 2024; Liu et al., 2023b; Yan et al., 2024). The first line of work distills the performance of long sampling trajectories into student models capable of generating high-quality images with significantly fewer steps (Yin et al., 2024a; Zhou et al., 2024a). The second emphasizes trajectory consistency, ensuring that arbitrary sample points along a trajectory converge to the same endpoint, or that a two-step consecutive jump remains consistent with a one-step direct jump (Kim et al., 2024; Luo et al., 2023). The third straightens curved flows through rectification and reflow operations (Esser et al., 2024; Wang et al., 2025a), thereby enabling fast sampling. Although these approaches make one-step generation feasible, they generally require substantial computational resources, carefully designed discretization curricula, and multi-stage training (Geng et al., 2025a;b). Furthermore, they lack the flexibility to achieve a smooth trade-off between image quality and the number of neural function evaluations (NFEs).

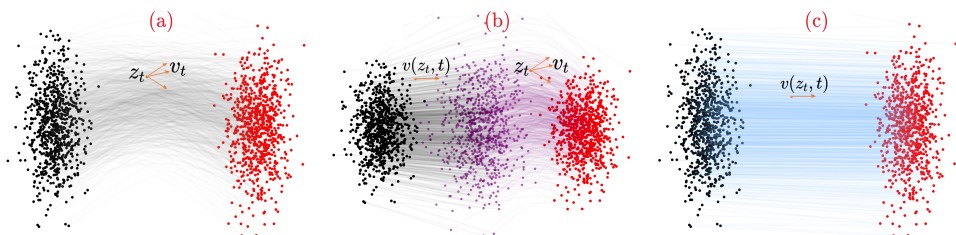

Figure 1: **Flow Uniqueness Models.** The standard idea in flow matching is to learn the velocity from randomly composed pairs, which consequently suffers from velocity ambiguity that may correspond to different trajectories, as illustrated in (a). Since this ambiguity is the root cause of the limitations in one-step generation, we propose a novel and effective framework, *FUM*, to address it. Concretely, the entire flow path is divided into two sub-paths, where the velocity of the first sub-path preserves strong uniqueness by being learned from strictly one-to-one pairs, as shown in (b). By further introducing a flow consistency strategy that enforces the velocity of the second sub-path to align with that of the first, the resulting sampling path naturally inherits this uniqueness, as seen in (c).

Recently, flow matching has shown promising potential for one-step generative modeling by learning a time-dependent marginal velocity field and leveraging the learned velocity to drive generation. However, since this velocity field is trained from randomly paired noise–image samples (Liu et al., 2023a; Lipman et al., 2023), it may be induced by multiple distinct pairings (Geng et al., 2025a), resulting in ambiguity in the learned velocity. To define a unique velocity field, another line of work employs a pre-trained DM to synthesize one-to-one corresponding noise–image pairs (Liu et al., 2023b). Yet this approach incurs substantial generation overhead and suffers from unavoidable truncation errors due to intrinsic limitations of DMs (Karras et al., 2022). As a result, both noise–image pair construction methods prevent flow matching from fully unleashing its modeling potential.

To remedy this, we put forward a simple and effective framework, termed *Flow Uniqueness Models* (**FUM**), as shown in Figure 1. The core idea of FUM is to build the uniqueness for the velocity field, preventing it from being shared across multiple noise–image pairs. In contrast to standard flow matching methods that achieve velocity uniqueness through multi-stage rectification and reflow operations (Lipman et al., 2023; Liu et al., 2023b), FUM constructs velocity uniqueness directly by leveraging strictly one-to-one image pairs. Importantly, FUM can be initialized from a pre-trained DM and further optimized with only modest computational overhead, or trained from scratch depending on the practical setting. By virtue of this modeling mechanism, FUM not only achieves high-quality one-step generation but also provides flexibility to trade off image quality against NFEs.

To this end, we divide the entire path between the prior and target distributions into two sub-paths, with the split point determined by the deterministic nature of ODE samplers (Song et al., 2021a; Lu et al., 2022). Specifically, when applied to a pre-trained DM, there exists a time point at which an ODE sampler can deterministically denoise a sample to the final endpoint (Kim et al., 2025b; 2022), yielding an image that is highly aligned with its ground-truth counterpart. Although the two may not perfectly coincide, this still establishes an implicit one-to-one correspondence. Leveraging this property, we construct image pairs via a time-interpolation operation (Albergo & Vanden-Eijnden, 2023; Lipman et al., 2023), where the interpolated time corresponds to the split point. Since the two images in each constructed pair are strictly one-to-one, the resulting velocity field is uniquely defined. Optimizing the first sub-path with these pairs thus yields a unique and reliable flow structure (Liu et al., 2023a), which serves as the cornerstone for few-step sampling. Moreover, unlike pairs generated by a pre-trained DM that suffer from unavoidable truncation errors, our constructed pairs lie on the ground-truth flow path and are therefore free from such errors. Consequently, the velocity on the first sub-path can be learned efficiently, analogous to reflow operations in standard flow matching (Liu et al., 2023b; Stoica et al., 2025). Nevertheless, the second sub-path cannot be optimized in the same manner, as corresponding one-to-one pairs are unavailable, and it must instead learn the velocity from randomly matched pairs (Liu et al., 2023a; Zhu et al., 2025).

To obtain a continuous sampling path with a unique velocity field, we design a flow-consistency strategy that aligns the velocity of the second sub-path with that of the first. Inspired by Shortcut models (Frans et al., 2025) and MeanFlow (Geng et al., 2025a), we incorporate their key principles into FUM by exploiting the additivity of integrals. Theoretically, the integral from a point on the first

sub-path to the split point, plus the integral from the split point to a point on the second sub-path, equals the integral from the first point directly to the second. Hence, since the first sub-path preserves velocity uniqueness, this additivity guarantees that the entire sampling path naturally inherits the same uniqueness. Building on this property, FUM yields an entire and continuous sampling path with a unique velocity field, thereby achieving remarkable one-step generative modeling.

In short, we summarize our contributions as follows: 1) We propose a novel and effective framework, FUM, which achieves impressive one-step generative performance while offering flexibility to trade off between image quality and sampling steps. 2) To enable this, we divide the entire flow path into two sub-paths and construct strictly one-to-one corresponding image pairs on the first sub-path, which allows us to enforce velocity uniqueness in this segment. 3) We introduce a flow-consistency strategy that aligns the velocity on the second sub-path with that of the first, yielding an entire sampling path that naturally inherits velocity uniqueness. 4) Extensive experiments on three benchmark datasets demonstrate the remarkable empirical performance of FUM.

## 2 PRELIMINARY:FLOW MATCHING

To situate our proposed FUM within the broader landscape, we begin by revisiting the foundational principles of flow matching principle, the framework that underpins our approach.

Flow matching is a recently developed family of generative models that learn to match flows, represented by velocity fields, between a target distribution $p_{\text{data}}(x_0)$ and a prior distribution $\pi = \mathcal{N}(0, \text{I})$. Formally, given data $x_0 \sim p_{\text{data}}(x_0)$ and prior noise $\epsilon \sim \pi$, a flow path can be defined as: $z_t = a_t x_0 + b_t \epsilon, t \in [0, 1]$, where $a_t$ and $b_t$ are predefined schedules satisfying boundary conditions, such as $a_0 = 1, b_0 = 0$ and $a_1 = 0, b_1 = 1$. A common and simple choice is the linear schedule $a_t = 1 - t$ and $b_t = t$, which defines a straight-line trajectory from at $t = 0$ to $\epsilon$ at $t = 1$. Associated with this path is the instantaneous velocity field $v_t$, defined as the time derivative of the flow path: $v_t = z_t' = a_t' x_0 + b_t' \epsilon$. Since this velocity is defined with respect to a specific data sample $x_0$, it is referred to as the conditional velocity (Lipman et al., 2023), denoted $v_t(z_t \mid x_0)$. For the linear schedule described above, the conditional velocity reduces to the simple form $v_t = \epsilon - x_0$.

**Training:** In practice, a given sample $z_t$ along a sampling trajectory could have been generated from many different noise–image pairs $(\epsilon, x_0)$ (Guo et al., 2025; Stoica et al., 2025). Therefore, the goal of a generative model is not to learn any single conditional velocity, but rather the average over all such possibilities—namely, the marginal velocity $v(z_t, t) = \mathbb{E}[v(z_t, t) \mid z_t]$. However, directly computing this marginal velocity and optimizing a loss against it is intractable. To address this, the flow-matching framework introduces an elegant and practical surrogate: the conditional flow-matching loss. This objective trains a neural network $v_\theta$ by minimizing the discrepancy between its prediction and an easily computable conditional velocity. Concretely, the loss is $\mathcal{L}_{\text{CFM}}(\theta) = \mathbb{E}_{t, x_0, \epsilon} \|v_\theta(z_t, t) - v(z_t, t)\|^2 = \mathbb{E}_{t, x_0, \epsilon} \|v_\theta(z_t, t) - (a_t' x_0 + b_t' \epsilon)\|^2$. It has been shown that minimizing this conditional objective is equivalent to minimizing the loss with respect to the true marginal velocity field. Consequently, by optimizing $\mathcal{L}_{\text{CFM}}$, the network $v_\theta$ learns the vector field that governs transport of the overall data distribution.

**Sampling:** Given a marginal velocity field $v_\theta(z_t, t)$, samples are generated by solving an ODE $dz_t = v(z_t, t)dt$, starting from $z_1 \sim \mathcal{N}(0, \text{I})$. The solution can be written as: $z_r = z_t - \int_t^r v(z_\tau, \tau)d\tau$, where $r < t$ refers to another time step in the same sampling trajectory. In practice, this integral is approximated numerically over discrete time steps. For example, using the optimized velocity model $v_{\theta^*}$, the Euler method (Karras et al., 2022), a first-order ODE solver, updates each step as $z_r = z_t + (r - t)v_{\theta^*}(z_t, t)$. Analogously, higher-order solvers can also be applied (Lu et al., 2022; Zhang & Chen, 2023) to solve the above ODE. However, generating high-quality samples $z_0$ requires multiple evaluation steps, since solving the integral lies in a high-dimensional space, which has motivated research into more efficient few-step and one-step generation methods.

## 3 FLOW UNIQUENESS MODELS

The core idea of FUM is to preserve the uniqueness of the velocity field $v_t$, preventing it from simultaneously representing multiple trajectories defined by different noise-image pairs. To this end, the entire flow path is divided into two sub-paths, with the velocity of the first sub-path determined

---

**Algorithm 1** FUM Training Procedure

**Require:** Dataset $\mathcal{D} = \{x_0\}$, prior $\pi$, reversible range $(0, R)$, velocity model $v_\theta$, learning rate $\eta$, EMA decay rate schedule $\mu(\cdot)$, consistency optimization variant (`Shortcut` or `MeanFlow`)

1: $\theta^- \leftarrow \theta$ and $k \leftarrow 0$
2: **while** not converged **do**
3:      Sample $x_0 \sim p_{\text{data}}$, $\epsilon \sim \mathcal{N}(0, \text{I})$, $s \sim \mathcal{U}(0, R)$
4:      Construct $x_s = \alpha(s)x_0 + \beta(s)\epsilon$
5:      Form strictly paired $(x_0, x_s)$ for $\text{P}(x_s \rightarrow x_0)$
6:      Form random pairs $(\epsilon, x_s)$ for $\text{P}(\epsilon \rightarrow x_s)$
7:      **if** `Shortcut` **then**
8:          Compute $v_{\text{target}} = \frac{1}{2}[v_\theta(x_i, i, s-i) + v_\theta(x_s, s, j-s)]$
9:          Compute $\mathcal{L}(\theta)$ using Eq. (2)
10:     **else if** `MeanFlow` **then**
11:         Compute $(v_{\text{fir\_target}}, v_{\text{sec\_target}}, v_{\text{target}})$ via JVP
12:         Compute $\mathcal{L}(\theta)$ using Eq. (3)
13:     **end if**
14:     $\theta \leftarrow \theta - \eta\nabla_\theta\mathcal{L}(\theta)$
15:     $\theta^- \leftarrow \text{stopgrad}(\mu(k)\theta^- + (1 - \mu(k))\theta)$, $k = k + 1$
16: **end while**

---

**Algorithm 2** Sampling with FUM

**Require:** Optimized velocity model $v_{\theta*}$, step size $\Delta k$ (control the sampling steps), consistency optimization variant (`Shortcut` or `MeanFlow`)

1: Initialize $x \sim \mathcal{N}(0, I)$, $k \leftarrow 0$
2: **for** $k \in [0, 1]$ **do**
3:      **if** `Shortcut` **then**
4:          $x \leftarrow x + \Delta k \cdot v_{\theta*}(x, k, \Delta k)$
5:      **else if** `MeanFlow` **then**
6:          $x \leftarrow x - \Delta k \cdot v_{\theta*}(x, k + \Delta k, k)$
7:      **end if**
8:      $k \leftarrow k + \Delta k$
9: **end for**
10: **Output:** Final image $x$

---

by strictly one-to-one image pairs, ensuring strong uniqueness. By introducing the flow consistency strategy, the velocity of the second sub-path is optimized to follow that of the first, yielding a complete sampling path that inherits the intrinsic uniqueness of the first sub-path. Building on this modeling philosophy, FUM not only achieves high-quality one-step generation but also provides the flexibility to trade off between image quality and the number of sampling steps.

Below, we first describe the flow path division for constructing strictly one-to-one pairs, then introduce the flow consistency strategy, followed by an analysis of its theoretical advantages, and finally present the model optimization techniques. When integrated into our framework, the training procedure is conceptually summarized in Algorithm 1, while sampling can be performed by Algorithm 2.

## 3.1 FLOW PATH DIVISION

The flow path is divided to construct strictly one-to-one image pairs, leveraging the deterministic nature of ODE-based samplers (Song et al., 2021a; Lu et al., 2022). Specifically, when an ODE sampler discretizes the continuous time horizon $t \in [0, 1]$, it produces a deterministic sampling trajectory $\mathcal{L} = \{x_{t_0}, ..., x_{t_s}, ..., x_{t_R}, ..., x_{t_N}\}$: once the initial sample $x_{t_N}$ is fixed, every intermediate state along $\mathcal{L}$ is uniquely determined. In theory, this determinism implies that any state $x_{t_s}$ on the trajectory can be reversed back to its endpoint $x_{t_0}$ (Song et al., 2023). In practice, however, ODE samplers are reliably reversible only over a limited prefix of the trajectory (Kim et al., 2025b; 2022), because numerical truncation errors accumulate. We denote the largest practically reversible time step by $t_R$, and define the reversible range as $\{x_{t_s}\}_{s\in[1,R]}$. Notably, determining $R$ incurs negligible overhead: for a fixed dataset and sampling schedule, $R$ is a dataset-level constant, so we do not need to search for a separate $R$ for each individual sample $x_0$. Within this range, each pair $(x_{t_0}, x_{t_s})$ forms an implicit one-to-one correspondence, as the ODE sampler can deterministically map $x_{t_s}$ to $\tilde{x}_{t_0}$ (Figure 6). Although $\tilde{x}_{t_0}$ may not exactly equal $x_{t_0}$, it remains sufficiently close in semantics to establish a reliable bijection, allowing us to treat $(x_{t_0}, x_{t_s})$ as strictly paired samples.

**First Sub-path** $\text{P}(x_s \rightarrow x_0)$**.** Motivated by the above insights, we split the continuous flow path from the prior distribution $\pi$ to the target data distribution $p_{\text{data}}(x_0)$ into two sub-paths at a reversible time $s \in (0, R]$, where $R$ corresponds to $t_R$ in $\mathcal{L}$. This introduces an intermediate distribution $p(x_s)$, which serves as the pivot between the two sub-paths. Consequently, we enable to construct image pairs $(x_0, x_s)$ through the forward diffusion:

$$x_s = \alpha(s)x_0 + \beta(s)\epsilon_s, \epsilon_s \sim \mathcal{N}(0, \text{I}), s \in (0, R]. \tag{1}$$

Here, $\alpha(s)$ and $\beta(s)$ are differentiable functions of $s$ with bounded derivatives, as detailed in (Song et al., 2021b; Lu et al., 2022). Because each $x_s$ is uniquely associated with its corresponding $x_0$, the

velocity defined between them is uniquely determined and cannot be realized by any other pair. For convenience, we denote the first sub-path as $P(x_s \to x_0)$, whose velocity field $v(z_i, i)$ is therefore one-to-one flow-unique for all $i \in (0, s)$. Moreover, for a given target dataset, once $s$ is determined, constructing image pairs via Eq. (1) incurs only minimal computational cost, since it requires no neural-network evaluations.

**Second Sub-path** $P(\epsilon \to x_s)$. Although the first sub-path $P(x_s \to x_0)$ is explicitly defined via strict one-to-one pairs, the second sub-path cannot be constructed in the same manner, because such correspondence is no longer guaranteed for times beyond $s$. Specifically, we sample noise $\epsilon$ from the prior $\pi$ and pair it with a randomly selected $x_s$, thereby defining the second sub-path $P(\epsilon \to x_s)$. In this region, the induced velocity field $v(z_j, j)$ on $P(\epsilon \to x_s)$ may correspond to multiple underlying flows and thus lacks intrinsic uniqueness for $j \in (s, 1)$. Therefore, the key challenge is to resolve this ambiguity and enforce a unique velocity field for $P(\epsilon \to x_s)$.

On the other hand, the ultimate goal of generative modeling is to traverse the sampling path from the prior noise $\epsilon$ to the data sample $x_0$ (Sabour et al., 2025; Kim et al., 2025a). In FUM, because we divide the overall flow into two sub-paths, we must combine $P(\epsilon \to x_s)$ and $P(x_s \to x_0)$ to recover the complete sampling path $P(\epsilon \to x_0)$. Hence, another key aspect of optimizing FUM is to ensure that the velocity field on $P(\epsilon \to x_s)$ transitions smoothly into that on $P(x_s \to x_0)$, thereby establishing a coherent and consistent flow over the entire path $P(\epsilon \to x_0)$.

### 3.2 Flow Consistency Strategy

To address the two challenges mentioned above, we introduce a flow-consistency strategy that enforces the velocity of $P(\epsilon \to x_s)$ to seamlessly follow that of $P(x_s \to x_0)$. This strategy is implemented in two variants, leveraging the modeling principles of Shortcut models (Frans et al., 2025) and MeanFlow (Geng et al., 2025a). Through this design, we obtain a complete and continuous sampling path $P(\epsilon \to x_0)$ with a unique velocity field—one that cannot be replicated by different noise-image pairs—and optimize it end-to-end.

The conceptual idea of our flow consistency strategy is built on the additivity property of the integral:

$$\int_i^j v(x_\tau, \tau)d\tau = \int_i^s v(x_\tau, \tau)d\tau + \int_s^j v(x_\tau, \tau)d\tau, s \in (i, j).$$

In other words, the integral over the complete path $P(\epsilon \to x_0)$, defined from $i$ to $j$, can be decomposed into the sum of the integrals over the two sub-paths: $P(x_s \to x_0)$ (from $i$ to $s$) and $P(\epsilon \to x_s)$ (from $s$ to $j$). Leveraging this relationship, the velocity field of $P(\epsilon \to x_s)$ can be naturally aligned with that of $P(x_s \to x_0)$, thereby yielding a complete sampling path $P(\epsilon \to x_0)$ with a smooth and consistent flow. Since the pairs $(x_0, x_s)$ are in strict one-to-one correspondence, matching the velocity field of $P(x_s \to x_0)$ ensures strong uniqueness, which then propagates to $P(\epsilon \to x_0)$ through the flow consistency strategy. Consequently, $P(\epsilon \to x_0)$ can be effectively employed for few-step generative modeling. In what follows, we describe how the design philosophies of Shortcut models (Frans et al., 2025) and MeanFlow (Geng et al., 2025a) are incorporated into our strategy.

**Shortcut Models-based Strategy.** The basic modeling idea of Shortcut models is to leverage an inherent self-consistency property, namely, that one shortcut step is equivalent to two consecutive shortcut steps of half the size:

$$u(x_i, i, j - i) = v(x_i, i, s - i)/2 + v(x_s, s, j - s)/2,$$

In contrast to the step-size constraint in Shortcut models, where $(s - i)$ must equal $(j - s)$, we design our self-consistency targets to accommodate an unconstrained step-size relationship, such that $(s - i)$ is not required to equal $(j - s)$. Moreover, this formulation also remains valid in the degenerate case when $s - i = j - s = 0$. Under this principle, we optimize the neural network $u_\theta$ to learn the velocity field using the following loss function:

$$\mathcal{L}(\theta) = \mathbb{E}_{x_0 \sim p_{\text{data}}(x_0), x_s \sim p(x_s), \epsilon \sim \pi} \left[ \|u_\theta(x_i, i, 0) - (x_0 - x_s)\|_2^2 + \|u_\theta(x_j, j, 0) - (x_s - \epsilon)\|_2^2 \right]$$

$$+ \lambda * \mathbb{E} \left[ \|u_\theta(x_i, i, j - i) - u_{\text{target}}\|_2^2 \right], i \sim \mathcal{U}(0, s), j \sim \mathcal{U}(s, 1), \quad (2)$$

where $u_{\text{target}} = u_\theta(x_i, i, s - i)/2 + u_\theta(x_s, s, j - s)/2$, and $\lambda$ denotes the hyperparameter that balances the loss terms. Notably, both $i$ and $j$ are sampled from the uniform distribution. Unlike

the standard Shortcut models, FUM does not require denoising $x_j$ to obtain $x_s$, because $x_s$ is pre-determined. Intuitively, the above objective simultaneously accounts for velocity learning in both sub-paths while maintaining flow consistency between them. This ensures that the optimized $v_{\theta*}$ yields a smooth and coherent velocity field across the entire sampling path.

**MeanFlow-based Strategy.** The modeling principle in MeanFlow is to introduce a new field that represents the average velocity. Formally, the average velocity representing the displacement between two time steps $i$ and $j$ can be defined as follows:

$$u(x_j, i, j) = \frac{1}{j-i} \int_j^i v(x_\tau, \tau) d\tau.$$

Similarly to Shortcut models, $u(x_j, i, j)$ is a velocity field jointly dependent on $(i, j)$. When incorporating MeanFlow into FUM optimization, our training objective can be formulated as:

$$\mathcal{L}(\theta) = \mathbb{E}_{x_0 \sim p_{\text{data}}(x_0), x_s \sim p(x_s), \epsilon \sim \pi} \left[ \|u_\theta(x_i, 0, i) - u_{\text{fir\_target}}\|_2^2 + \|u_\theta(x_j, s, j) - u_{\text{sec\_target}}\|_2^2 \right]$$
$$+ \mathbb{E} \left[ \|u_\theta(x_j, i, j) - u_{\text{target}}\|_2^2 \right], i \sim \mathcal{U}(0, s), j \sim \mathcal{U}(s, 1), \quad (3)$$

where $u_{\text{fir\_target}}$, $u_{\text{sec\_target}}$, and $u_{\text{target}}$ are derived using the JVP operation (Geng et al., 2025a). For simplicity, we provide only their definitions below, with detailed derivations are deferred to the appendix:

$$u_{\text{fir\_target}} = v_s - (s-i) \cdot \Big( v_s\, \partial_x u_\theta(x_s, i, s) + \partial_s u_\theta(x_s, i, s) \Big),$$

$$u_{\text{sec\_target}} = v_j - (j-s) \cdot \Big( v_j\, \partial_x u_\theta(x_j, s, j) + \partial_j u_\theta(x_j, s, j) \Big),$$

$$u_{\text{target}} = v_j - (j-i) \cdot \Big( v_j\, \partial_x u_\theta(x_j, i, j) + \partial_j u_\theta(x_j, i, j) \Big).$$

Since $s$ in FUM is treated as a constant, we have $\partial_s u_\theta(x_s, i, s) = 0$, which reduces the first target to $u_{\text{fir\_target}} = v_s - (s-i) \cdot v_s \partial_x u_\theta(x_s, i, s)$. Intuitively, the objective serves the same role as Eq. (2): it learns the average velocity for both sub-paths and directly enforces the average velocity in $\mathrm{P}(\epsilon \to x_s)$ to follow that in $\mathrm{P}(x_s \to x_0)$. Consequently, the optimized $u_{\theta*}$ simultaneously captures the velocities of both sub-paths while preserving flow consistency between them, thereby yielding a complete sampling path with strong uniqueness.

Building on the above flow consistency strategy, our FUM effectively addresses the two aforementioned challenges, ensuring uniqueness in the second sub-path and enforcing its velocity consistency with that of the first. Moreover, we validate FUM through experiments that implement the flow consistency strategy in both variants, each consistently demonstrating strong effectiveness.

### 3.3 THEORETICAL ANALYSIS

In this section, we mainly provide a theoretical analysis of our proposed FUM, showing how velocity uniqueness and flow consistency are preserved across the complete sampling path.

As shown in Eq. (1), constructing $(x_0, x_s)$ through the forward diffusion guarantees a deterministic mapping between $x_0$ and $x_s$, thereby preserving strong uniqueness in the velocity field. Formally:

**Proposition 1** (Velocity Uniqueness in the First Sub-path). *For any $s \in (0, R]$, the mapping $x_0 \mapsto x_s$ defined by Eq. (1) is injective almost surely. Therefore, the velocity $v(x_s, s)$ along $\mathrm{P}(x_s \to x_0)$ is unique and cannot be reproduced by any other pair $(x_0', x_s')$.*

*Proof sketch.* Assume that two distinct images sampled from the target dataset, $x_0 \neq x_0'$, produce the same $x_s$ via Eq. (1). Then $\alpha(s)x_0 + \beta(s)\epsilon_s = \alpha(s)x_0' + \beta(s)\epsilon_s$, which implies $\Rightarrow \alpha(s)(x_0 - x_0') = 0$. Since $\alpha(s) \neq 0$ on $(0, R]$, we must have $x_0 = x_0'$, contradicting the assumption that $x_0 \neq x_0'$. Hence, the mapping is injective, and $(x_0, x_s)$ forms a strict one-to-one correspondence. $\qquad \square$

This eliminates conditional ambiguity on $\mathrm{P}(x_s \to x_0)$: every intermediate state on the reversible segment has a unique pre-image, so the marginal velocity equals the unique conditional velocity (up to a small tolerance induced by numerical truncation). As a result, optimizing with strict pairs yields an intrinsically unique velocity field on the first sub-path. However, practical generation begins

from $\epsilon$ and terminates at $x_0$, so we must ensure that the velocity of $\mathrm{P}(\epsilon \to x_s)$ aligns with that of $\mathrm{P}(x_s \to x_0)$. Following our method, we leverage additivity of line integrals, for any $i < s < j$,

$$\int_i^j v(z_\tau, \tau) d\tau = \int_i^s v(z_\tau, \tau) d\tau + \int_s^j v(z_\tau, \tau) d\tau.$$

**Proposition 2** (Inherited Uniqueness from the First Sub-path). *Assuming $\mathrm{P}(x_s \to x_0)$ induces a unique velocity field and the velocity on $\mathrm{P}(\epsilon \to x_s)$ is constrained to match it via our flow-consistency strategy, the additivity of integrals guarantees that $\mathrm{P}(\epsilon \to x_0)$ follows the same unique continuation, yielding a continuous, flow-unique velocity field over the entire horizon.*

*Proof sketch.* Let $z_t^*$ denote the unique marginal velocity field on $t \in [0, s]$, and and let it be extended to $t > s$ by the ground-truth marginal velocity. Our flow-consistency strategy enforces that the displacement between any sampled pair $(i, j)$ decomposes additively, thereby anchoring the split condition at $t = s$ to be consistent with $P(x_s \to x_0)$. Since an ODE driven by a Lipschitz velocity field admits a unique solution given any split state, the sub-path $P(\epsilon \to x_s)$ can only continue from $P(x_s \to x_0)$ along the unique continuation determined by $P(x_s \to x_0)$. Consequently, the entire sampling path $P(\epsilon \to x_0)$ inherits the velocity-uniqueness of $P(x_s \to x_0)$. $\square$

Based on the above analysis, our proposed FUM enjoys two key guarantees: 1) **Velocity Uniqueness Guarantee.** The strictly paired $(x_0, x_s)$ enforce a unique velocity field on $\mathrm{P}(x_s \to x_0)$, and this uniqueness propagates to the entire path via Proposition 2; 2) **Flow Consistency Guarantee.** By the additivity of integrals, enforcing velocity alignment at the split time $s$ yields global flow consistency, preventing mismatches between the two sub-paths. Together, these guarantees explain why FUM can achieve high-quality one-step generation while supporting flexible trade-offs between image quality and sampling steps. Specifically, velocity uniqueness eliminates multi-flow ambiguity, and the flow-consistency constraint ensures globally correct transport along the entire sampling path.

### 3.4 OPTIMIZATION DETAILS

To effectively optimize $v_\theta$, we incorporate advanced training techniques to enhance its modeling capacity, including data augmentation, stable training strategy, and dynamic time selection.

**Data Augmentation.** Practically, data capacity plays a key role in fully optimizing the velocity model. Hence, in our FUM training, we employ the H-Flip (Zhu et al., 2025) operation to double the coupled pairs. For instance, given an image pair $(x_0, x_s)$, both images can be horizontally flipped, resulting in a new paired sample $(\mathrm{H} - \mathrm{Flip}(x_0), \mathrm{H} - \mathrm{Flip}(x_s))$. Moreover, during training, we sample a batch of $M$ image pairs for the first sub-path and another batch of $M$ pairs for the second sub-path, where the $x_s$ in $(x_0, x_s)$ directly corresponds to the $x_s$ in $(x_s, \epsilon)$.

**Stable Training.** Current advanced DMs often employ an exponential moving average (EMA) (Karras et al., 2022; Song et al., 2023) over weight parameters to enhance modeling capacity. Concretely, EMA introduces a smoothing effect on generations, which is particularly beneficial in diffusion modeling since the training objective inherently exhibits high variance. Following this setting, we also adopt EMA in optimizing FUM. This is motivated by the fact that the loss for enforcing flow consistency between the two sub-paths introduces substantial oscillations. By utilizing EMA parameters to generate self-consistency targets, this issue can be effectively alleviated.

**Dynamic Time Selection.** In our FUM optimization, three distinct time values, $i$, $s$, and $j$, must be selected prior to each training iteration. For the path division time $s$, its maximum reversible value is denoted as $R$, and $i$ must always be smaller than $s$. Given a specific dataset and the corresponding pre-trained DM, we first determine the exact value of $R$ for the their default ODE sampler. Based on this, we sample $s$ from the uniform distribution $s \sim \mathcal{U}(0, R)$, followed by sampling $i$ from $i \sim \mathcal{U}(0, s)$. For $j$, since it does not exhibit an explicit dependency on $i$ or $s$, we sample it independently from the uniform distribution $j \sim \mathcal{U}(s, 1)$.

## 4 EXPERIMENTS

This section presents the experimental results of FUM. We begin by describing the training implementation details, followed by comparisons with advanced approaches and comprehensive ablation studies. In addition, qualitative results are presented in Figure 3 and Figure 5, respectively.

Table 1: **Unconditional Results on CIFAR-10.** Table 2: **Conditional Generative Performance Evaluation on ImageNet** $64 * 64$**.**

| Model | FID↓ | NFE↓ |
|---|---|---|
| **Diffusion + Distillation** | | |
| PD (Salimans & Ho, 2022) | 9.12 | 1 |
| PD | 4.51 | 2 |
| CD (Song et al., 2023) | 3.55 | 1 |
| CD | 2.93 | 2 |
| sCD (Lu & Song, 2025) | 3.66 | 1 |
| sCD | 2.52 | 2 |
| **Consistency Models** | | |
| CT (Song et al., 2023) | 8.70 | 1 |
| CT | 5.83 | 2 |
| iCT (Song & Dhariwal, 2024) | 2.83 | 1 |
| iCT | 2.46 | 2 |
| sCT (Lu & Song, 2025) | 2.85 | 1 |
| sCT | 2.06 | 2 |
| ECM (Geng et al., 2025b) | 3.60 | 1 |
| ECM | 2.11 | 2 |
| CTM (Kim et al., 2024) | 5.19 | 1 |
| CTM+GAN (Kim et al., 2024) | **1.87** | 2 |
| CTM+GAN | 1.98 | 1 |
| **Flow Matching** | | |
| 1-rectified flow (Liu, 2022) | 6.18 | 1 |
| 2-rectified flow | 4.85 | 1 |
| 3-rectified flow | 5.21 | 1 |
| 2-rectified flow++ (Lee et al., 2024) | 3.07 | 1 |
| 2-rectified flow++ | 2.40 | 2 |
| SlimFlow (Zhu et al., 2025) | 4.53 | 1 |
| IMM (Zhou et al., 2025) | 3.20 | 1 |
| IMM | 1.98 | 2 |
| MeanFlow (Geng et al., 2025a) | 2.92 | 1 |
| FUM(ours)+Shortcut consistency | **1.93** | 4 |
| FUM(ours)+Shortcut consistency | **1.92** | 2 |
| FUM(ours)+Shortcut consistency | **2.01** | 1 |
| FUM(ours)+Meanflow consistency | **1.99** | 4 |
| FUM(ours)+Meanflow consistency | **2.04** | 2 |
| FUM(ours)+Meanflow consistency | **2.10** | 1 |

| Model | FID↓ | NFE↓ |
|---|---|---|
| **Diffusion + Distillation** | | |
| PD (Salimans & Ho, 2022) | 15.39 | 1 |
| PD | 8.95 | 2 |
| PD | 6.77 | 4 |
| CD (Song et al., 2023) | 6.20 | 1 |
| CD | 4.70 | 2 |
| sCD (Lu & Song, 2025) | 2.44 | 1 |
| sCD | 1.66 | 2 |
| **Consistency Models** | | |
| CT (Song et al., 2023) | 13.0 | 1 |
| CT | 11.1 | 2 |
| iCT (Song & Dhariwal, 2024) | 4.02 | 1 |
| iCT | 3.20 | 2 |
| iCT-deep (Song & Dhariwal, 2024) | 3.25 | 1 |
| iCT-deep | 2.77 | 2 |
| sCT (Lu & Song, 2025) | 2.04 | 1 |
| sCT | 2.06 | 1 |
| ECM-S (Geng et al., 2025b) | 4.05 | 1 |
| ECM-S | 2.79 | 2 |
| ECM-XL (Geng et al., 2025b) | 2.49 | 1 |
| ECM-XL | **1.67** | 2 |
| CTM+GAN (Kim et al., 2024) | 1.92 | 1 |
| CTM+GAN | 1.73 | 2 |
| **Flow Matching** | | |
| 2-rectified flow++ (Lee et al., 2024) | 4.31 | 1 |
| 2-rectified flow++ | 3.64 | 2 |
| SlimFlow (Zhu et al., 2025) | 12.34 | 1 |
| FUM(ours)+Shortcut consistency | **2.36** | 4 |
| FUM(ours)+Shortcut consistency | **2.71** | 2 |
| FUM(ours)+Shortcut consistency | **2.96** | 1 |
| FUM(ours)+MeanFlow consistency | **2.27** | 4 |
| FUM(ours)+MeanFlow consistency | **2.35** | 2 |
| FUM(ours)+MeanFlow consistency | **2.54** | 1 |

## 4.1 IMPLEMENTATION DETAILS

We evaluate our FUM framework on three widely-used benchmark datasets: CIFAR-10, FFHQ-$64 \times 64$, and ImageNet at two resolutions, $64 \times 64$ and $256 \times 256$, following established conventions for experimental setups and evaluation metrics. During training, we initialize FUM with pre-trained DMs, applying only minor modifications. This design choice primarily aims to preserve velocity uniqueness through strictly one-to-one image pairs, which are obtained from the deterministic nature of the ODE sampler in the pre-trained model. Specifically, we employ the pre-trained EDM (Karras et al., 2022) for CIFAR-10, ImageNet-64, and FFHQ-64, while for ImageNet-256, we utilize the pre-trained DiT (Peebles & Xie, 2023). Notably, we also train FUM from scratch to further evaluate its universality, with details provided in the Appendix. When implementing the two proposed flow consistency strategies, we optimize FUM with two distinct variants, both of which demonstrate strong performance. To validate this, we report FID scores, NFEs, and model parameters as quantitative evidence of FUM's superiority. Additional training details, including batch size, training iterations, and hyperparameter settings, are provided in the Appendix.

## 4.2 PERFORMANCE COMPARISON

**One-step Generator.** Our main contribution lies in demonstrating the advanced performance of one-step generation. To this end, we conduct experiments with two flow consistency strategies, Shortcut-based and MeanFlow-based, across three commonly used datasets. Results are presented in Table 1, Table 3, Table 2, and Table 4. For instance, as shown in Table 1, FUM achieves state-of-the-art (SoTA) performance under both both strategies. Although CTM+GAN (Kim et al., 2024) attains the best FID, it relies on a discriminator, introducing potential instability that our framework avoids. On ImageNet-256, FUM achieves results comparable to the SoTA MeanFlow (Geng et al., 2025a), as shown in Table 4, further highlighting its effectiveness and robustness.

Table 3: **Performance on FFHQ** $64 \times 64$.

| Model | FID↓ | NFE↓ |
|---|---|---|
| **Diffusion** | | |
| EDM (VP) (Karras et al., 2022) | **2.39** | 79 |
| EDM (VE) | 2.53 | 79 |
| **Diffusion + Fast Samplers** | | |
| DDIM (Song et al., 2021a) | 18.30 | 10 |
| AMED-Solver (Zhou et al., 2024b) | 6.31 | 9 |
| AMED-Plugin (Zhou et al., 2024b) | 4.24 | 9 |
| **Flow Matching** | | |
| SlimFlow (Zhu et al., 2025) | 7.21 | 1 |
| 2-rectified flow++ (Lee et al., 2024) | 5.21 | 1 |
| 2-rectified flow++ | 4.26 | 2 |
| FUM(ours)+Shortcut consistency | **3.27** | 4 |
| FUM(ours)+Shortcut consistency | **3.40** | 2 |
| FUM(ours)+Shortcut consistency | **3.72** | 1 |
| FUM(ours)+MeanFlow consistency | **3.35** | 4 |
| FUM(ours)+MeanFlow consistency | **3.46** | 2 |
| FUM(ours)+MeanFlow consistency | **3.73** | 1 |

Table 4: **Performance on ImageNet** $256 \times 256$.

| Model | params↓ | FID↓ | NFE↓ |
|---|---|---|---|
| **Diffusion Models** | | | |
| DiT-XL/2 (Peebles & Xie, 2023) | 675M | **2.27** | $2 * 250$ |
| SiT-XL/2 (Ma et al., 2024) | 675M | **2.06** | $2 * 250$ |
| **Consistency Models** | | | |
| iCT-XL/2 (Song & Dhariwal, 2024) | 675M | 34.24 | 1 |
| iCT-XL/2 | 675M | 20.30 | 2 |
| **Flow Matching** | | | |
| iMM-XL/2 (Zhou et al., 2025) | 675M | 7.77 | $2 * 1$ |
| Shortcut-XL/2 (Frans et al., 2025) | 675M | 10.60 | 1 |
| MeanFlow-XL/2+ (Geng et al., 2025a) | 676M | **2.20** | 2 |
| MeanFlow-XL/2 | 676M | 2.93 | 2 |
| MeanFlow-XL/2 | 676M | 3.43 | 1 |
| FUM(ours)+Shortcut consistency | 675M | **2.41** | 4 |
| FUM(ours)+Shortcut consistency | 675M | **2.62** | 2 |
| FUM(ours)+Shortcut consistency | 675M | **3.23** | 1 |
| FUM(ours)+MeanFlow consistency | 675M | **3.32** | 4 |
| FUM(ours)+MeanFlow consistency | 675M | **4.17** | 2 |
| FUM(ours)+MeanFlow consistency | 675M | **5.10** | 1 |

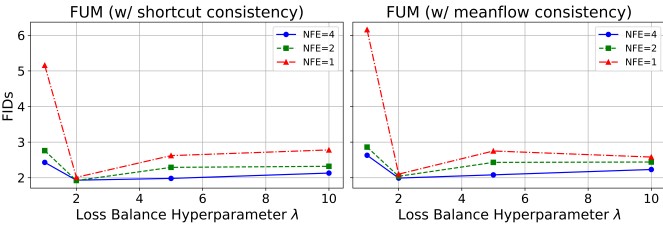

Figure 2: **Ablation of Hyperparameter** $\lambda$ **on CIFAR-10.**

Table 5: **Ablation of Optimization Techniques on CIFAR-10.**

| Technique | FID↓ | NFEs |
|---|---|---|
| **shortcut consistency** | | |
| +H-Flip | 2.14 | 1 |
| +H-Flip+EMA | 2.01 | 1 |
| **meanflow consistency** | | |
| +H-Flip | 2.23 | 1 |
| +H-Flip+EMA | 2.10 | 1 |

**Few-step Trade-off.** Compared with previous one-step generators, our FUM demonstrates a strong ability to flexibly balance image quality and the number of NFEs. As shown in Table 1, Table 3, Table 2, and Table 4, FUM consistently improves performance when additional NFEs are utilized. For example, in Table 4, FUM outperforms MeanFlow with 4 and 2 NFEs, while falling slightly behind with a single NFE. These results clearly verify the capacity of the optimized FUM to achieve a smooth trade-off, a property absent in previous advanced one-step generators.

### 4.3 ABLATION STUDY

We conduct comprehensive ablation studies to validate the design choices of our FUM framework. Specifically, we first evaluate the effect of the hyperparameter $\lambda$ on CIFAR-10, as shown in Figure 2. In addition, we examine the optimization technique used to enhance FUM, with results reported in Table 5. Together, these findings clearly demonstrate the effectiveness of our design.

## 5 CONCLUSION

In this paper, we propose a novel and effective framework, named **FUM**, which not only achieves high-quality one-step generation but also enables flexible trade-offs between image quality and sampling steps. Broadly speaking, the success of this framework stems from its deterministic construction of strictly one-to-one corresponding image pairs, which naturally preserves velocity-field uniqueness. Since the entire flow path is divided into two sub-paths to obtain these pairs, this property is limited to the first sub-path and requires considerable effort to enforce alignment of the velocity in the second sub-path with that of the first. To overcome this limitation, we introduce a flow consistency strategy, comprising two distinct variants, that produces a complete sampling path with consistent flow, thereby inheriting the uniqueness from the first sub-path. Building on this design philosophy, FUM presents remarkable empirical performance across three benchmark datasets.

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

## A    Using LLM for Polishing Writing

We leverage a large language model (LLM) to assist with paper writing, including refining wording, enhancing clarity, and performing preliminary proofreading. The LLM streamlines the editing process by suggesting improved phrasing and correcting minor grammatical errors, allowing us to focus on the technical content. Since this contributes to improving readability and maintaining a clear logical flow, we disclose it here in accordance with the ICLR policy.

## B    Limitations

Although our proposed FUM demonstrates remarkable generative performance, certain issues still limit the full realization of its superiority. First, in practice, there remains a possibility that our method may produce images with noticeable artifacts. Second, compared with MeanFlow (Geng et al., 2025a), Shortcut Models (Frans et al., 2025), and IMM (Zhou et al., 2025), FUM is not a purely one-stage training scheme, as it relies on pre-trained DMs to obtain strictly one-to-one pairs through the deterministic nature of ODEs. In this context, although our framework achieves superior results in one-step generation, there remains room for further enhancement, such as improving sampling stability and enabling a purely one-stage training paradigm.

## C    Related Work

**Diffusion Models.**  DMs have long been recognized as a powerful class of generative models, achieving state-of-the-art (SoTA) performance in image (Karras et al., 2024), audio (Majumder et al., 2024), video (Chen et al., 2024; Wang et al., 2024b), and 3D generation tasks (Go et al., 2025; Lin et al., 2023). A large body of work has focused on improving their sampling efficiency, as the original iterative denoising process typically requires hundreds or thousands of NFEs (Lu et al., 2022; Song et al., 2021a; Zhao et al., 2023). To address this bottleneck, subsequent research has proposed advanced samplers (Zhang & Chen, 2023; Tong et al., 2025; Zhou et al., 2024b; Sabour et al., 2024), distillation techniques (Sauer et al., 2024a;b), and consistency strategies (Song et al., 2023; Kim et al., 2024) that significantly reduce the NFEs while maintaining generation quality.

**Flow Matching.**  Flow matching has recently emerged as an alternative to DMs by reformulating generative modeling as learning deterministic continuous flows between the prior and data distributions (Lipman et al., 2023; Ma et al., 2024; Boffi et al., 2025), thereby eliminating stochastic noise perturbations and instead relying on velocity fields (Albergo & Vanden-Eijnden, 2023). Subsequent works have advanced flow matching with improved solvers and well-designed training objectives, achieving faster convergence and more stable learning dynamics (Peng et al., 2025; Stoica et al., 2025; Tong et al., 2024). Rectified flow (Liu et al., 2023a), a variant of flow matching, has further refined this paradigm by explicitly straightening sampling trajectories (Ma et al., 2025), thereby improving both efficiency and effectiveness (Esser et al., 2024; Liu et al., 2023b; Wang et al., 2025a).

**One-step Generator.**  Since iterative sampling limits the practical employment of generative models, numerous efforts have been devoted to developing advanced one-step generators. For instance, consistency-based approaches (Song & Dhariwal, 2024; Wang et al., 2024a; Kim et al., 2024; Geng et al., 2025b; Lu & Song, 2025) achieve strong one-step generative performance but require substantial computational resources to maintain trajectory consistency. Distillation-based methods (Yin et al., 2024b;a; Sauer et al., 2024a; Salimans et al., 2024) further enhance modeling performance by incorporating discriminators into the training process. In contrast, the flow-matching paradigm, including IMM (Zhou et al., 2025), Shortcut models (Frans et al., 2025), and MeanFlow (Geng et al., 2025a), offers a complementary perspective for advancing one-step generation. Despite these advances, achieving high-quality one-step generation remains an open challenge, motivating research into new frameworks that enforce trajectory or flow consistency to further expand modeling capacity.

## D    Detailed Derivation of the JVP-based Targets

We now present the detailed JVP derivation of the three target velocities, $v_{\text{fir\_target}}$, $v_{\text{sec\_target}}$, $v_{\text{target}}$, employed in the MeanFlow-based consistency strategy for optimizing FUM.

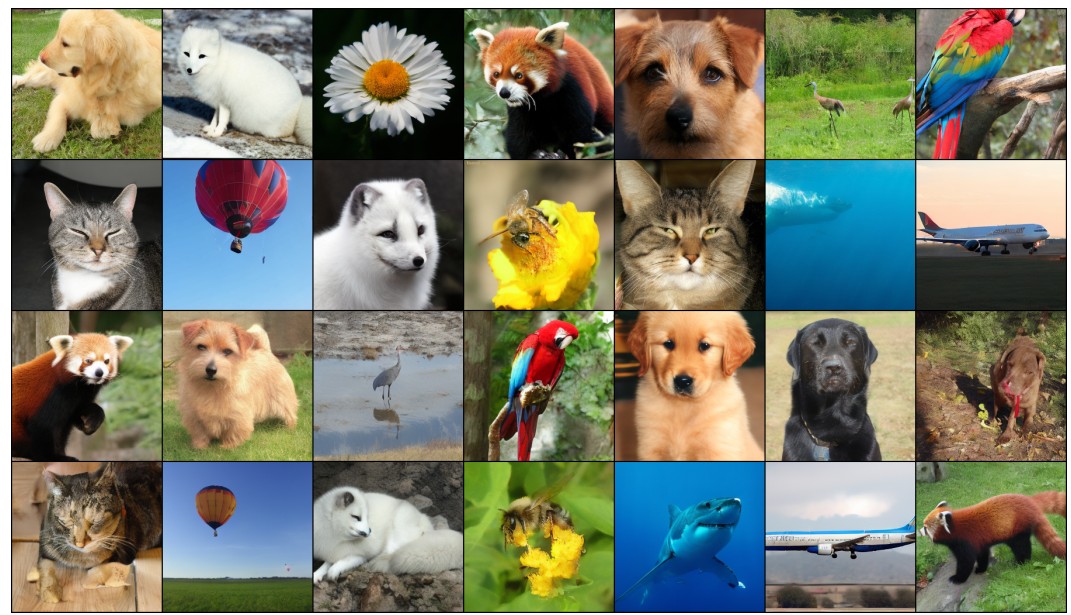

Figure 3: **Randomly Generated Samples on ImageNet-**$256 \times 256$ **via 1-NFE Inference**.

**General Taylor Expansion.** Consider two times $s$ and $j = s + \Delta$ with states $x_s, x_j$. By first-order expansion, we have

$$x_j \approx x_s + (j - s)\, v_s, \quad v_s := v(x_s, s).$$

Expanding $v(x_j, j)$ around $(x_s, s)$ yields

$$v(x_j, j) \approx v(x_s, s) + \partial_x v(x_s, s) \cdot (x_j - x_s) + \partial_t v(x_s, s)(j - s).$$

Substituting $x_j - x_s \approx (j - s)v_s$ gives

$$v(x_j, j) \approx v_s + (j - s)\Big(\partial_x v(x_s, s) \cdot v_s + \partial_t v(x_s, s)\Big).$$

The term $\partial_x v(x_s, s) \cdot v_s$ is exactly the *Jacobian–Vector Product (JVP)*:

$$\mathrm{JVP}(v; x_s, v_s) = \nabla_x v(x_s, s) \cdot v_s.$$

Rearranging gives the template relation:

$$v_s \approx v(x_j, j) - (j - s)\Big(\mathrm{JVP}(v; x_s, v_s) + \partial_t v(x_s, s)\Big).$$

**First Sub-path Target** $v_{\mathrm{fir\_target}}$**.** For the sub-path $\mathrm{P}(x_s \to x_0)$ with $i < s$, we expand $v(x_s, s)$ around $(x_i, i)$:

$$v(x_s, s) \approx v(x_i, i) + (s - i)\Big(\mathrm{JVP}(v; x_s, v_s) + \partial_s v(x_s, i, s)\Big).$$

Rearranging gives

$$v_{\mathrm{fir\_target}} = v_s - (s - i)\Big(v_s\, \partial_x v_\theta(x_s, i, s) + \partial_s v_\theta(x_s, i, s)\Big).$$

Since $s$ is fixed in FUM, we have $\partial_s v_\theta(x_s, i, s) = 0$, so

$$v_{\mathrm{fir\_target}} = v_s - (s - i)\, v_s\, \partial_x v_\theta(x_s, i, s).$$

**Second Sub-path Target** $v_{\mathrm{sec\_target}}$**.** For the sub-path $\mathrm{P}(\epsilon \to x_s)$ with $s < j$, we expand $v(x_j, j)$ around $(x_s, s)$:

$$v(x_j, j) \approx v(x_s, s) + (j - s)\Big(\mathrm{JVP}(v; x_j, v_j) + \partial_j v(x_j, s, j)\Big).$$

Table 6: Training Settings for FUM Optimization.

| Pre-trained Baselines | Dataset | Consistency Strategy | Training Iterations | Batch Size | FIDs | NFEs |
|---|---|---|---|---|---|---|
| EDM | CIFAR-10 | Shortcut | 60K | 256 | 2.11 | 4 |
| EDM | CIFAR-10 | Shortcut | 60K | 256 | 2.34 | 2 |
| EDM | CIFAR-10 | Shortcut | 60K | 256 | 2.99 | 1 |
| EDM | CIFAR-10 | MeanFlow | 55K | 256 | 2.10 | 4 |
| EDM | CIFAR-10 | MeanFlow | 55K | 256 | 2.35 | 2 |
| EDM | CIFAR-10 | MeanFlow | 55K | 256 | 2.78 | 1 |
| EDM | CIFAR-10 | Shortcut | 40K | 512 | 1.93 | 4 |
| EDM | CIFAR-10 | Shortcut | 40K | 512 | 1.92 | 2 |
| EDM | CIFAR-10 | Shortcut | 40K | 512 | 2.01 | 1 |
| EDM | CIFAR-10 | MeanFlow | 40K | 512 | 1.99 | 4 |
| EDM | CIFAR-10 | MeanFlow | 40K | 512 | 2.04 | 2 |
| EDM | CIFAR-10 | MeanFlow | 40K | 512 | 2.10 | 1 |
| EDM | FFHQ-64 | Shortcut | 40K | 512 | 3.27 | 4 |
| EDM | FFHQ-64 | Shortcut | 40K | 512 | 3.40 | 2 |
| EDM | FFHQ-64 | Shortcut | 40K | 512 | 3.72 | 1 |
| EDM | FFHQ-64 | MeanFlow | 40K | 512 | 3.35 | 4 |
| EDM | FFHQ-64 | MeanFlow | 40K | 512 | 3.46 | 2 |
| EDM | FFHQ-64 | MeanFlow | 40K | 512 | 3.73 | 1 |
| EDM | ImageNet-64 | Shortcut | 50K | 1024 | 2.36 | 4 |
| EDM | ImageNet-64 | Shortcut | 50K | 1024 | 2.71 | 2 |
| EDM | ImageNet-64 | Shortcut | 50K | 1024 | 2.96 | 1 |
| EDM | ImageNet-64 | MeanFlow | 50K | 1024 | 2.27 | 4 |
| EDM | ImageNet-64 | MeanFlow | 50K | 1024 | 2.35 | 2 |
| EDM | ImageNet-64 | MeanFlow | 50K | 1024 | 2.54 | 1 |
| DiT-XL/2 | ImageNet-256 | Shortcut | 65K | 256 | 2.41 | 4 |
| DiT-XL/2 | ImageNet-256 | Shortcut | 65K | 256 | 2.62 | 2 |
| DiT-XL/2 | ImageNet-256 | Shortcut | 65K | 256 | 3.23 | 1 |
| DiT-XL/2 | ImageNet-256 | MeanFlow | 60K | 256 | 3.32 | 4 |
| DiT-XL/2 | ImageNet-256 | MeanFlow | 60K | 256 | 4.17 | 2 |
| DiT-XL/2 | ImageNet-256 | MeanFlow | 60K | 256 | 5.10 | 1 |

Rearranging gives

$$v_{\mathrm{sec\_target}} = v_j - (j - s)\Big(v_j\,\partial_x v_\theta(x_j, s, j) + \partial_j v_\theta(x_j, s, j)\Big).$$

**Complete-Path Target** $v_{\mathrm{target}}$. For the entire path $\mathrm{P}(\epsilon \to x_0)$ with $i < s < j$, we expand $v(x_j, j)$ around $(x_i, i)$:

$$v(x_j, j) \approx v(x_i, i) + (j - i)\Big(\mathrm{JVP}(v; x_j, v_j) + \partial_j v(x_j, i, j)\Big).$$

Rearranging gives

$$v_{\mathrm{target}} = v_j - (j - i)\Big(v_j\,\partial_x v_\theta(x_j, i, j) + \partial_j v_\theta(x_j, i, j)\Big).$$

**Consolidated Results.** Summarizing, the three JVP-based targets are

$$v_{\mathrm{fir\_target}} = v_s - (s - i) \cdot \Big(v_s\,\partial_x v_\theta(x_s, i, s) + \partial_s v_\theta(x_s, i, s)\Big),$$

$$v_{\mathrm{sec\_target}} = v_j - (j - s) \cdot \Big(v_j\,\partial_x v_\theta(x_j, s, j) + \partial_j v_\theta(x_j, s, j)\Big),$$

$$v_{\mathrm{target}} = v_j - (j - i) \cdot \Big(v_j\,\partial_x v_\theta(x_j, i, j) + \partial_j v_\theta(x_j, i, j)\Big).$$

In summary, these targets arise directly from a Taylor expansion of the MeanFlow average velocity and ensure that the velocity of the second sub-path aligns with the velocity field of the first sub-path, which is learned from strictly one-to-one image pairs. Consequently, the complete sampling path naturally inherits the uniqueness property.

Table 7: Training FUM from Scratch.

| Models | FID ↓ | NFE ↓ | Training Iterations | Batch Size |
|---|---|---|---|---|
| IMM (Zhou et al., 2025) | 3.20 | 1 | 400k | 4096 |
| MeanFlow (Geng et al., 2025a) | 2.92 | 1 | 800k | 1024 |
| Ours (Based on pretrained EDM) | **2.01** | 1 | 400k (EDM) + 40k (FUM) | 512 |
| Ours (Training from Scratch) | 2.69 | 1 | 750k | 512 |
| Ours (Training from Scratch) | 2.64 | 1 | 800k | 512 |
| Ours (Training from Scratch) | 2.53 | 1 | 600k | 1024 |
| Ours (Training from Scratch) | 2.41 | 1 | 800k | 1024 |

Table 8: Performance on Text-to-Image Generation.

| Models | FID-5k ↓ | CLIP ↑ | NFE |
|---|---|---|---|
| SD (Rombach et al., 2022) | 20.1 | 0.318 | 25 |
| InstaFlow (Liu et al., 2023b) | 23.4 | 0.304 | 1 |
| **Ours** | **22.5** | **0.311** | **1** |

# E  MORE EXPERIMENT DETAILS

In this section, we provide further details on our experiments for optimizing FUM, which are crucial for demonstrating its effectiveness and extensibility. Since FUM is initialized with pre-trained baselines, we directly adopt their hyperparameter settings, including the learning rate, EMA decay, and model optimizer. We also report the training iterations, batch size, and the corresponding FIDs under this setting, as summarized in Table 6. In addition, we present further results on convergence performance to more comprehensively illustrate the optimization process, as detailed in Figure 4.

**Training FUM from Scratch**. Moreover, to verify that FUM can also be trained from scratch, we conducted preliminary experiments on CIFAR-10 with a randomly initialized EDM backbone. As the detailed results are reported in Table 7, FUM consistently achieves competitive one-step performance when trained from scratch, which further supports its generality. We believe this setting can be improved with more specialized end-to-end training strategies, which we leave for future work.

Table 10: Split Point Sensitivity Testing based on ImageNet-256.

| $t$ | 7 | 9 | 11 | 13 | 15 | 17 |
|---|---|---|---|---|---|---|
| FUM + Shortcut consistency | 10.64 | 6.70 | 5.19 | 3.87 | **3.23** | 80.36 |

**Sensitivity Testing**. To test sensitivity to the split point, we conduct additional ablations on ImageNet-256 using a pre-trained DiT-XL/2 (Peebles & Xie, 2023). We discretize the sampling horizon into 50 EDM scheduler steps (Karras et al., 2022). Empirically, we find that $t = 15$ is the largest split point at which the sampler can still denoise $x_t$ back to $\tilde{x}_0$ while preserving high semantic consistency with $x_0$ (Kim et al., 2025b). We therefore respectively test $t \in \{7, 9, 11, 13, 15, 17\}$ and report one-step (1 NFE) FID results in Table 10. Clearly, the best generation performance is achieved at $t = 15$. Within the feasible range $t \leq 15$, performance degrades gradually as $t$ decreases. We attribute this trend to the fact that maintaining one-to-one uniqueness over a longer early segment yields smoother and more reliable velocity consistency over the full path. When $t = 17$, performance drops sharply because the base sampler can no longer reliably denoise $x_t$ to an output that remains semantically similar to $x_0$, violating the prerequisite for constructing valid one-to-one pairs. Hence, based on the empirical analysis above, it is reasonable to set the largest feasible split point as $R$ in our main paper and conduct training using Algorithm 1. Concretely, once $R$ is determined, we uniformly sample $s$ from $(0, R]$ before each training iteration and construct the corresponding image pair $(x_0, x_s)$ to build the velocity uniqueness.

Table 9: Image Diversity Testing.

| Models | FID ↓ | NFE | Precision ↑ | Recall ↑ |
|---|---|---|---|---|
| DiT-XL/2 (Peebles & Xie, 2023) | 2.27 | 2×250 | 0.67 | 0.67 |
| Ours + Shortcut consistency | 3.23 | 1 | 0.63 | 0.65 |
| Ours + MeanFlow consistency | 5.10 | 1 | 0.64 | 0.66 |

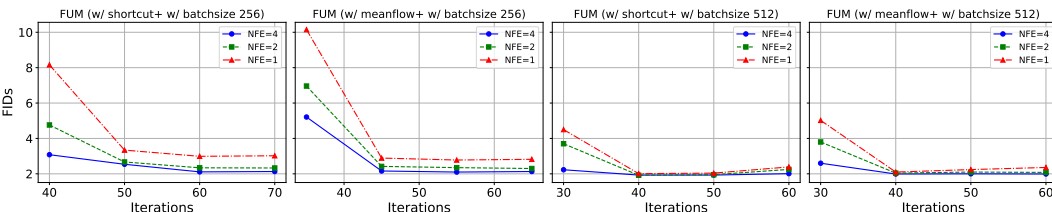

Figure 4: **Experimental Results under Different Training Iterations on CIFAR-10**.

**Text-to-Image Generation**. Since multimodal generation is central to current AI research, we conduct preliminary experiments on MS COCO 2017 using a pre-trained Stable Diffusion (SD) model (Rombach et al., 2022) to further validate the robustness of FUM. We also compare against a representative one-step text-to-image baseline, InstaFlow (Liu et al., 2023b), which is built on the flow-matching framework. As shown in Table 8, the results demonstrate that FUM can be effectively extended to text-to-image generation. We view this as a promising first step and plan to explore larger-scale text-conditioned backbones and broader multimodal settings (e.g., text-to-video and other cross-modal generation tasks) in future work.

**Image Diversity Testing**. To evaluate whether strict one-to-one pairing harms diversity, we report Precision and Recall on ImageNet-256, which jointly reflect sample fidelity and diversity. As shown in Table 9, both Precision and Recall remain comparable to the baseline, with no significant drop in Recall, indicating that our framework does not noticeably reduce sample diversity. Moreover, we do not observe systematic visual artifacts in the generated samples (Figure 3). Therefore, strict one-to-one pairing improves one-step modeling without sacrificing diversity.

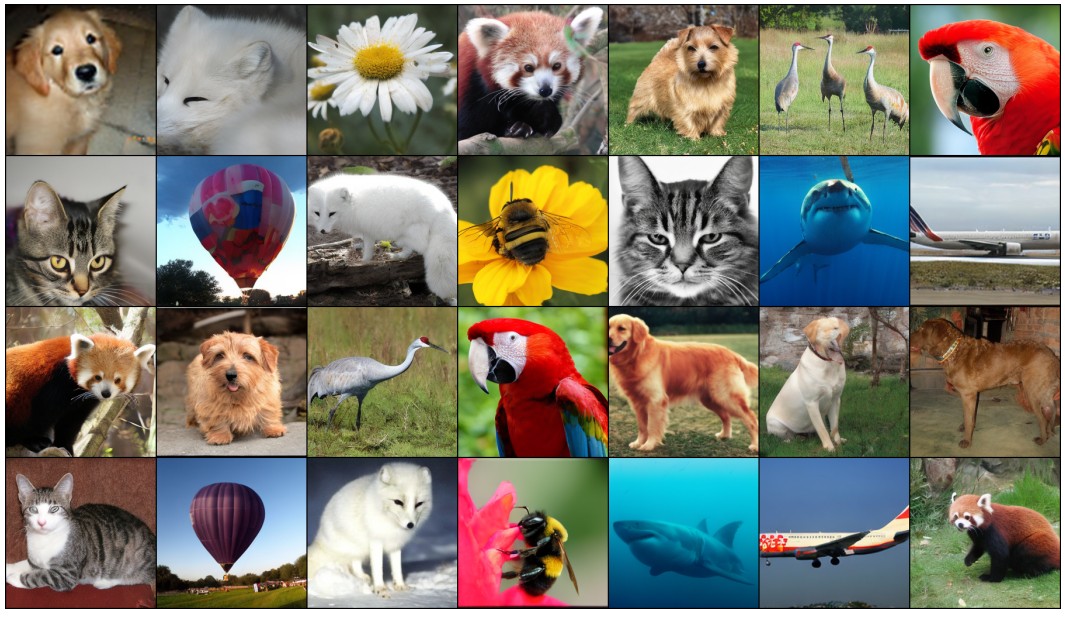

Figure 5: **Randomly Generated Samples on ImageNet-**$256 \times 256$ **with 8-NFE**.

$x_0$ 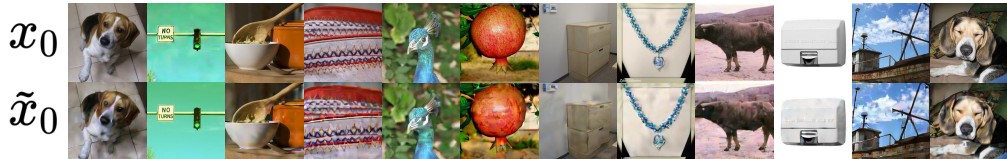

$\tilde{x}_0$

Figure 6: **Deterministic Nature of ODE Samplers: An Illustration.** When a data sample $x_0 \sim p_{\mathrm{data}}(x_0)$ is perturbed to $x_s$, an ODE sampler can deterministically denoise $x_s$ back to $\tilde{x}_0$, where $x_0$ and $\tilde{x}_0$ still share highly similar structural and semantic content. Importantly, $x_0$ cannot be obtained from any other intermediate sample $x_s'$ that is distinct from $x_s$, because the mapping within the reversible range is strictly one-to-one. In this context, we construct the image pair $(x_0, x_s)$, which induces a velocity field with strong (*i.e.*, one-to-one) uniqueness.

