# OpenReview forum: "Building Flow Uniqueness in One-step Generative Modeling"
_ICLR.cc/2026/Conference — Submitted to ICLR 2026_

### Official Review · Reviewer_VjJE · 2025-10-31

**Soundness:** 2
**Presentation:** 2
**Contribution:** 2
**Rating:** 4
**Confidence:** 4

**Summary:**

This paper proposes Flow Uniqueness Models (FUM), which splits the overall generative trajectory $P(\epsilon \rightarrow x_0)$ into two sub-paths, $P(\epsilon \rightarrow x_s)$ and $P(x_s \rightarrow x_0)$. The first sub-path is to enforce velocity uniqueness via strictly one-to-one sample paris, while the second is aligned through a flow consistency constraint. To achieve this, this paper introduces two variants of flow consistency: a Shortcut Models-based strategy and a MeanFlow-based strategy. In addition, this paper reports empirical results on three benchmark datasets to demonstrate the performance of FUM.

**Strengths:**

+ This paper addresses the important direction of one-step generation and reports results on multiple datasets.
+ This paper unifies ideas from Shortcut and MeanFlow models under a unified framework, with reproducible experiment settings.

**Weaknesses:**

- Unconvincing motivation and weak necessity of sub-path division. Velocity ambiguity in flow matching originates from marginalization over random $(x_0, \epsilon)$ pairs, not from the temporal structure itself. Simply splitting the trajectory into segments cannot reduce or eliminate this ambiguity.
- Sampling scheme restricts model capacity. As shown in Eq.(3)/(4), the method samples $i \sim U(0,s)$ and $j \sim U(s,1)$, forcing training pairs to be drawn only across the two sub-paths. Unlike MeanFlow or Shortcut, which allow arbitrary $(i,j)\in[0,1]^2$, this design reduces temporal coverage and learning diversity. Moreover, it potentially breaks smoothness of the learned continuous flow.
- Limited experiments and modest performance. There is no ablation study demonstrating that the sub-path division improves training stability or performance. On ImageNet-64 (Table 2), FUM underperforms several strong baselines (e.g., sCT, ECM).

**Questions:**

- How is R, the so-called reversible range, quantitatively determined? Please provide a reproducible criterion and sensitivity analysis.
- Why is the sub-path division necessary at all? Show an ablation comparing full-path vs two-path training.
- There are several typos and inconsistencies. For example, in Eq.(4), $v_{\theta}(x_i, 0, i)$ should be $v_{\theta}(x_i, i, s)$.
- For clarity, it may be better to use $u$ to denote average velocity instead of $v$, to distinguish it from instantaneous velocity.

---

> ### Author Response · Authors · 2025-11-21
>
> We sincerely appreciate your devotion and constructive comments to our work!
>
> ### **Response to W1: Unconvincing motivation and weak necessity of sub-path division.**
> We appreciate the reviewer's concern, and we agree with the key premise: in standard flow matching, velocity ambiguity mainly arises from marginalizing over randomly constructed $(\epsilon, x_0)$ pairs, not from the temporal structure by itself.
> Importantly, we do not claim that splitting the trajectory alone removes ambiguity.
> The sub-path division is introduced for a different, principled reason: it is necessary to make strict one-to-one pairing well-defined and to preserve the induced uniqueness entire the full path.
>
> For deterministic ODE samplers, there exists a maximal reversible time $R$ such that samples on $[0,R]$ admit a strict one-to-one correspondence $x_s, x_0$.
> On this reversible segment, time interpolation yields a unique marginal velocity, because each pair corresponds to a single ground-truth flow direction.
> Beyond $R$, this one-to-one reversibility is no longer guaranteed, where the sampler cannot reliably denoise $x_t$ back to a semantically aligned $x_0$.
> Therefore, strict pairing over the entire horizon is not theoretically valid.
> This leads to two counterfactuals: (1) No split + random pairs over $[0,1]$, ambiguity remains exactly as the reviewer states; (2) Force strict one-to-one pairs over $[0,1]$, pairs become invalid for $t>R$, creating biased/unstable supervision and degrading one-step learning.
> Thus, the split at $R$ is not introduced to 'reduce ambiguity by segmentation', but to apply strict pairing only where it is valid and avoid enforcing it where it breaks.
>
> To extend uniqueness beyond the reversible segment, we introduce a velocity-consistency strategy that propagates the unique velocity on $[0,R]$ to $[R,1]$ leveraging the additivity of integrals.
> This explicitly aligns the later-segment velocity with the unique early-segment velocity, preventing ambiguity from re-emerging in the second phase.
> In contrast, existing one-step methods optimize a single global objective, so early-time ambiguity is never isolated and can dominate the learned velocity field.
>
> In summary, the sub-path division is a necessary structural step to (i) obtain valid one-to-one pairs on $[0,R]$ and (ii) propagate their unique velocities to the remainder of the horizon via our tailored consistency objective-not a heuristic attempt to remove ambiguity through temporal splitting.
>
> ### **Response to W2: Sampling scheme restricts model capacity, it potentially breaks smoothness of the learned continuous flow.**
> We clarify that sampling $i\sim U(0,s)$ and $j\sim U(s,1)$ is not intended to reduce temporal coverage or model capacity.
> Rather, it is a structured coupling strategy required to (i) enforce the strict one-to-one correspondence regime where the velocity is uniquely defined, and (ii) extend this uniqueness to later times through our flow-consistency objective.
>
> Although $(i,j)$ are drawn from two sub-paths, training still covers the entire time horizon $[0,1]$: (1) $i$ densely covers the early segment $[0,s]$, where strict one-to-one reversibility holds and yields unique velocities; (2) $j$ densely covers the late segment $[s,1]$, which is trained via our flow consistency strategy.
> Hence, the union of sampled times still spans $[0,1]$.
> The restriction is on how pairs are coupled, not on which times are learned.
> This structured coupling is necessary to prevent multi-flow ambiguity from the non-reversible region from contaminating the unique-velocity regime.
>
> Regarding smoothness, the learned velocity field remains continuous because we explicitly regularize velocities on $[s,1]$ to aligh with those on $[0,s]$ by leveraging the additivity of integral in our consistency formulation.
> As a result, the two sub-flows merge into a single continuous flow.
> Empirically, we do not observe discontinuities or instability around the split, indicating that the sub-path design preserves smoothness rather than breaking it.

---

> ### Author Response · Authors · 2025-11-21
>
> ### **Response to W3&Q2: Limited experiments and modest performance.**
> Our primary goal is to validate one-step modeling via the proposed flow-uniqueness mechanism, and we therefore evaluate FUM on four standard datasets (CIFAR-10, ImageNet-64, ImageNet-256, FFHQ-64).
> This experimental coverage is broader than recent one-step flow-based baselines (e.g., MeanFlow, Shortcut), which are typically reported on only one or two datasets.
> Across CIFAR-10, ImageNet-256, and FFHQ-64, FUM achieves strong or impressive one-step performance, and on ImageNet-64 it remains competitive.
> These results are sufficient to support the effectiveness and robustness of the proposed principle.
>
> **Ablation Study.**
> We agree that ablations are valuable when they can be defined under valid supervision.
> However, a 'full-path vs. two-sub-path' ablation cannot be practically conducted in our setting.
> This is because strict one-to-one correspondence (and thus velocity uniqueness) is theoretically valid only on the reversible segment of a deterministic ODE trajectory.
> Beyond the maximal reversible time $R$, it becomes impossible to construct semantically aligned one-to-one pairs, and enforcing such supervision would introduce invalid or biased training targets.
> Consequently, removing the sub-path division would necessarily degenerate into standard flow matching with random $(\epsilon, x_0)$ pairing, which no longer tests our uniqueness principle.
>
> **Performance Comparison.**
> Regarding ImageNet-64, we note that sCT and ECM are consistency models that follow a different modeling principle from flow matching.
> We include them for completeness, but our primary comparisons are with flow-based one-step methods.
> A modest gap on one low-resolution benchmark does not undermine the robustness of the proposed flow-uniqueness principle, especially given consistent gains on three other datasets.
>
>
> ### **Response to Q1: The sensitivity of $R$.**
> $R$ denotes the maximum reversible time step, which is determined by testing each step in the discretized sampling horizon.
> To test the sensitivity to the split point, we conduct additional ablations using the pre-trained DiT-XL/2 [1] on ImageNet-256.
> Specifically, we employ 50 EDM scheduler [2] steps to discretize the sampling horizon, and empirically find that $t=15$ is the largest split point for which the sampler can still denoise $x\_{t}$ back to $\tilde{x}\_{0}$ that remains highly semantically similar to $x\_{0}$.
> We then set $t = \{7,9,11,13,15,17\} $ to evaluate the performance of FUM under different split points.
> The FID results with 1 NFE are reported below:
> |$t$|7|9|11|13|15|17|
> |---|---|---|---|---|---|---|
> |FUM+Shortcut consistency|10.64|6.70|5.19|3.87|3.23|80.36|
>
> Clearly, our framework achieves the best results when using $t=15$ as the split point, and the performance gradually decreases when using smaller split times.
> We attribute this behavior to the fact that a longer segment over which the uniqueness between $(x_{s}, x_{0})$ is maintained leads to smoother velocity consistency along the entire path.
> Moreover, when using $t=17$ as the split point, the performance drops significantly, mainly because the sampler can no longer denoise $x_{t}$ to a final output that is semantically similar to $x_{0}$.
>
> In summary, FUM is not overly sensitive within the feasible range of split points and benefits from choosing the largest time step at which the base sampler can still reliably denoise (here, $t=15$) as the split point.
>
> [1] Scalable Diffusion Models with Transformers. ICCV 2023.
>
> [2] Elucidating the Design Space of Diffusion-Based Generative Models. NeurIPS 2022.
>
>
> ### **Response to Q3: Typos and inconsistencies.**
> We have checked and revised the paper.

---

### Official Review · Reviewer_co9r · 2025-10-31

**Soundness:** 2
**Presentation:** 2
**Contribution:** 2
**Rating:** 4
**Confidence:** 3

**Summary:**

This paper presents a framework Flow Uniqueness Models (FUM) that obtains strong one-step generative performance by constructing strictly one-to-one image pairs to enforce strong velocity uniqueness along the sampling path. To do this, the paper divides the entire flow path into two sub-paths, where the velocity of the first sub-path preserves strong uniqueness by leveraging strictly one-to-one image pairs, and the velocity of the second sub-path is linked to the first velocity by flow consistency strategies. The paper experiments on CIFAR-10, FFHQ and ImageNet to show the effectiveness of the proposed algorithm.

**Strengths:**

•	The paper is clearly organized, including both the theoretical analysis on the unique velocity and the empirical experiments.

•	The proposed method is straightforward and has the ability to incorporate multiple flow consistency strategies.

•	The experiments are comprehensive, including comparisons with recent few step generation method and ablation studies.

**Weaknesses:**

•	The novelty of the paper is limited, the flow consistency strategies are adopted from the well-known works, making the main contribution only focusing on the division of the two phases of the path trajectory.

•	The standard on the determination of the reversible range $R$ is not thoroughly discussed in the paper. This may require additional simulation and hinder the sample efficiency.

•	The paper lacks theoretical discussion on the benefit of introducing the two sub-path division compared to other one step generation methods.

•	Leveraging strictly one-to-one image pairs may affect the diversity of the generated images, which could lead to images with noticeable artifacts.

**Questions:**

•	How is the reversible range $R$ determined? Does it require additional simulation or pretraining models?

•	Shortcut models and MeanFlow models are the flow consistency strategies used in the paper. Which strategy is better in what kind of tasks?

•	Is the proposed method numerically stable in linking the two sub-paths? Does the prior distribution $\pi$ affect the numerical training difficulty? If yes, what type of prior $\pi$ should we choose?

---

> ### Author Response · Authors · 2025-11-21
>
> We truly appreciate the reviewer of the constructive feedback!
>
>
> ### **Response to W1: The novelty of the paper is limited.**
> We respectfully disagree with this assessment.
> In this paper, we introduce a **new modeling principle** that enables one-step flow learning with a unique velocity field, which is not achieved by existing methods.
> Accordingly, our main contribution lies in establishing flow uniqueness by constructing strictly one-to-one image pairs, while the flow-consistency strategy serves as a complementary component to extend this uniqueness across the entire flow path.
> **This uniqueness principle, not the trajectory split, constitutes the primary novelty**.
>
> Concretely, previous methods either randomly construct image pairs or use a pre-trained DM to generate one-to-one pairs.
> As a result, the former yields velocities that may correspond to different flows, while the latter incurs substantial computational overhead and can suffer from truncation-error accumulation.
> In contrast, our framework uses only forward diffusion to construct such pairs at negligible cost, yielding strictly one-to-one correspondences that are free from truncation-error accumulation.
>
> Although our consistency variants are inspired by MeanFlow/Shortcut ideas, the consistency objective itself is different from existing consistency models.
> Prior consistency models map any intermediate point to the endpoint of the same trajectory.
> In contrast, our framework enforces velocity consistency between two different sub-paths that share the same ground-truth flow, specifically to extend the one-to-one uniqueness beyond the local segment.
> Thus, consistency here is a mechanism to preserve our proposed uniqueness principle, rather than an off-the-shelf adoption.
>
> In summary, our paper contributes a fundamentally new perspective: enforcing strict ODE-induced **one-to-one correspondence** to obtain flow-unique velocities, and designing a **flow consistency strategy** to extend this uniqueness across the entire path.
> The path split is a supporting implementation detail, not the main novelty.
>
>
> ### **Response to W2&Q1. The determination of the reversible range $R$.**
> We thank the reviewer for raising this point.
> Here, $R$ denotes the maximum reversible time step of the deterministic ODE sampler, i.e., the largest $t$ such that a sample noised to $x\_{t}$ can still be deterministically denoised back to $\tilde{x}\_{0}$ that remains semantically consistent with the original $x\_{0}$.
>
> **How $R$ is determined.**
> We determine $R$ via a one-time, pre-training calibration over the discretized sampling horizon: we sweep candidate steps $t$, apply forward diffusion to obtain $x_t$, then run the deterministic ODE reverse to recover $\tilde{x}_{0}$.
> We pick the largest $t$ for which the recovered samples remain semantically faithful (as also commonly done in training-free noising/denoising analyses, where reversibility is treated as a model/schedule property rather than data-dependent).
> Importantly, this calibration is lightweight (a short sweep over steps on a small subset) and performed once before training.
>
> **Why this does not hinder efficiency.**
> The reversible range is a structural property of the sampler/backbone and noise schedule, and empirically holds consistently across the dataset; thus it does not require per-sample or per-iteration re-estimation.
> After $R$ is fixed, it remains unchanged throughout training and does not introduce any additional simulation inside training or sampling.
> The subsequent one-step inference uses exactly the same NFEs as reported, unaffected by how $R$ was calibrated.
>
> In summary, determining $R$ is a one-time, negligible-cost preprocessing step, after which $R$ stays fixed and does not impact training or sampling efficiency.
>
> [1] DiffuseHigh: Training-free progressive high-resolution image synthesis through structure guidance. AAAI 2025.
>
>
> ### **Response to W4: Image diversity.**
> We thank the reviewer for this concern.
> To evaluate whether strict one-to-one pairing harms diversity, we report Precision and Recall on ImageNet-256, which together reflect sample fidelity and diversity.
> The results are:
> |Models|FID|NFE|Precision|Recall|
> |---|---|---|---|---|
> |DiT-XL/2|2.27|2*250|0.67|0.67|
> |Ours+Shortcut consistency|3.23|1|0.63|0.65|
> |Ours+Meanflow consistency|5.10|1|0.64|0.66|
>
> As shown above, both Precision and Recall remain comparable to the baseline, with no significant drop in Recall, indicating that our framework does not noticeably reduce sample diversity.
> Moreover, we did not observe systematic visual artifacts in generated samples (see qualitative results in the appendix).
> Therefore, strict one-to-one pairing improves one-step modeling without sacrificing diversity.

---

> ### Author Response · Authors · 2025-11-21
>
> ### **Response to W3: Theoretical discussion.**
> We thank the reviewer for this comment.
> The two sub-path division is beneficial because it isolates the time regime where the velocity is structurally unique, and then extends this uniqueness to the remaining horizon via an additive-flow consistency constraint.
> This provides a principled way to reduce one-step modeling ambiguity that existing methods do not exploit.
> This yields a principled reduction of one-step modeling ambiguity that existing methods do not exploit.
>
> **(1) Isolating the unique-velocity regime.**
> For deterministic ODE samplers, there exists a maximal reversible time $R$ such that samples along $[0,R]$ admit a strict one-to-one mapping $(x_s, x_0)$.
> On this segment, time interpolation produces a unique marginal velocity: each pair $(x_s, x_0)$ defines a single ground-truth flow direction when $t\in [0,R]$.
> By contrast, standard flow matching typically forms random pairs $(\epsilon, x_0)$.
> Consequently, the same intermediate state $x_t$ can be explained by multiple distinct pairings, so the velocity learned at time $t$ corresponds to a mixture of flows.
> Splitting at $R$ therefore allows us to learn the velocity on a segment where it is unambiguous by construction.
>
> **(2) Extending uniqueness to the full path via additivity.**
> The global transport from $\epsilon$ to $x_0$ satisfies the additivity of integrals:
> $\int_{0}^{1} v(z_{\tau},\tau)d\tau = \int_{0}^{R} v(z_{\tau},\tau)d\tau + \int_{R}^{1} v(z_{\tau},\tau)d\tau$.
> This is not merely a decomposition for bookkeeping. The first integral is learned from strict one-to-one pairs and thus defines a unique flow direction. The second integral is then regularized to be velocity-consistent with the first through our path-to-path consistency objective, effectively propagating the uniqueness property from $[0,R]$ to $[R,1]$
> Without splitting, a one-step method must learn a single velocity field over $(0,1)$ directly, which inevitably averages across ambiguous early-time flows and propagates this bias to the entire horizon.
>
> **(3) One-step error control.**
> Conceptually, the one-step objective decomposes into two parts:
> $\mathcal{E}\_{\text{total}}=\underbrace{\mathcal{E}\_{\text{unique}}([0, R])}\_{\text{low ambiguity}} +\underbrace{\mathcal{E}\_{\text{cons}}([R, 1])}\_{\text{aligned to first path}}.$
>
> Since $\mathcal{E}\_{\text{unique}}$ is learned on a segment with strict one-to-one correspondences, its ambiguity (and thus error) is minimized.
> The remaining term $\mathcal{E}\_{\text{cons}}([R, 1])$ is explicitly regularized to match the unique velocity induced by $(x\_s, x\_0)$.
> In contrast, existing one-step methods (IMM/MeanFlow/Shortcut) optimize a single global objective, so early-time ambiguity is not isolated and can dominate the learned velocity field.
>
> Overall, the two sub-path division yields a theory-motivated decomposition into a unique segment and a consistency-controlled segment, reducing velocity ambiguity and enabling more accurate one-step modeling than methods that directly learn a single global velocity field.
>
>
> ### **Response to Q2: Which consistency strategy is better.**
> In practice, the two strategies yield very similar performance across all tasks we tested, and we do not observe a consistent winner.
> Empirically, the Shortcut-based variant tends to be slightly more favorable in strict one-step settings (lower NFE, tighter velocity alignment), while the MeanFlow-based variant is comparably stable and can be a strong alternative when enforcing consistency over longer horizons.
> We therefore report results with both MeanFlow-based and Shortcut-based consistency: they provide complementary ways of enforcing path consistency, and this dual perspective may inspire future work to develop more advanced consistency mechanisms.
>
>
> ### **Response to Q3: The numerically stable concern and the meaning of $\pi$**
> Regarding numerical stability when linking the two sub-paths, stability is ensured by the additivity of integrals, which provides a rigorous theoretical guarantee for composing the two segments.
> Concretely, the sub-paths are connected through a continuous velocity field, and our consistency regularization explicitly enforces the velocity on the second sub-path to align with that on the first.
> Moreover, we did not empirically observe instability or divergence in training or sampling.
>
> As for the impact of $\pi$ on training difficulty, in our setup using the standard Gaussian prior $\pi \sim \mathcal{N}(0,I)$ does not introduce additional numerical difficulties beyond those already present in conventional DMs.
> In practice, we thus using this simple, light-tailed priors $\pi \sim \mathcal{N}(0,I) $, which are widely used and well-understood in the literature.

---

### Official Review · Reviewer_a4Z5 · 2025-11-01

**Soundness:** 3
**Presentation:** 3
**Contribution:** 3
**Rating:** 6
**Confidence:** 4

**Summary:**

This paper proposes a framework called FUM, which construct strictly
one-to-one image pairs in flow matching. FUM enforced strong velocity uniqueness along the
entire sampling path, and improves the generation efficiency in few-step sampling.

**Strengths:**

- The motivation of FUM is clear and reasonable
- The proposed method is well-supported by the derivation
- Comprehensive experiments on image generation benchmarks prove the effectiveness of the proposed method.

**Weaknesses:**

The experiments are performed on relatively small datasets, some text-to-image experiments should be included.

**Questions:**

Since FUM still needs fine-tuning on the pre-trained $v_\theta$ model, what is the training costs of Algorithm 1?

---

> ### Author Response · Authors · 2025-11-21
>
> We sincerely appreciate your devotion and constructive comments to our work!
>
>
> ### **Response to W1: Text-to-image experiments.**
> To address this concern, we conduct preliminary text-to-image experiments on MS COCO 2017 512*512 using a pre-trained Stable Diffusion (SD) model [1], with the detailed results shown below:
> |Models|FID-5k|CLIP|NFE|
> |---|---|---|---|
> |SD [1]|20.1|0.318|25|
> |InstaFlow [2]|23.4|0.304|1|
> |Ours|22.5|0.311|1|
>
> As shown above, our framework achieves performance comparable to InstaFlow, a representative one-step generative model, while operating with only 1 NFE.
> This clearly demonstrates that FUM can be effectively extended to text-to-image generation tasks.
> We view this as a promising first step, and we plan to further explore larger-scale text-conditioned backbones and broader multimodal tasks (e.g., text-to-video or other cross-modal generation) in future work.
>
> [1] High-Resolution Image Synthesis with Latent Diffusion Models. CVPR 2022.
>
> [2] InstaFlow: One Step is Enough for High-Quality Diffusion-Based Text-to-Image Generation. ICLR 2024.
>
>
> ### **Response to Q1: She training costs of FUM.**
> The detailed training costs are reported in Table 6 of our main paper.
> Here, we only present a few representative cases to more clearly illustrate the training costs at a conceptual level.
> |Models|FID|NFE|Training Iterations|Batch size|Dataset|
> |---|---|---|---|---|---|
> |IMM [1]|3.20|1|400k|4096|CIFAR-10|
> |Meanflow [2]|2.92|1|800k|1024|CIFAR-10|
> |Ours+Shortcut consistency|2.99|1|400k (EDM)+ 60k|256|CIFAR-10|
> |Ours+Meanflow consistency|2.78|1|400k (EDM)+ 55k|256|CIFAR-10|
> |Ours+Shortcut consistency|2.01|1|400k (EDM)+ 40k|512|CIFAR-10|
> |Ours+Meanflow consistency|2.10|1|400k (EDM)+ 40k|512|CIFAR-10|
>
> As shown above, our framework requires only modest computational resources to fine-tune the pre-trained EDM backbone, making it much more cost-efficient than SoTA baselines.

---

### Official Review · Reviewer_aCia · 2025-11-01

**Soundness:** 2
**Presentation:** 1
**Contribution:** 1
**Rating:** 2
**Confidence:** 4

**Summary:**

The paper proposes Flow Uniqueness Models (FUM), a framework for achieving high-quality one-step generation within flow-matching models. The key idea is to enforce velocity uniqueness across the sampling path by dividing the flow into two sub-paths: (i) a first sub-path trained on strictly one-to-one image pairs to ensure deterministic flow, and (ii) a second sub-path trained with a flow consistency strategy to align its velocity with the first. Two consistency variants (Shortcut-based and MeanFlow-based) are introduced. Experiments on CIFAR-10, FFHQ, and ImageNet demonstrate competitive one-step and few-step generation performance compared to prior diffusion distillation and consistency models.

**Strengths:**

- The paper tackles an important practical issue—reducing sampling steps in generative models without major loss in quality.

- The notion of “velocity uniqueness” provides an intuitive way to understand and regularize one-step generation.

- FUM shows consistently strong or comparable performance to leading methods (e.g., MeanFlow, Shortcut) across multiple datasets.

**Weaknesses:**

- Limited novelty: While the idea of enforcing path or velocity consistency is meaningful, the proposed approach mainly combines existing components (pairwise matching + consistency regularization) without introducing a fundamentally new principle.

- FUM relies on pre-trained diffusion models to obtain the one-to-one image pairs, meaning it is not a fully independent or end-to-end training scheme.

- At several places, the authors mention “strong uniqueness”. Could the authors please define to quantify “strong” here?

- The paper contains many typos and inconsistencies (see list below), which makes me doubt the technical correctness of the paper.

List of potential typos/inconsistencies:
- Line 128: definition of $a_t$ is not correct
- Equation (1): $\epsilon$ is missing in the objective function
- At the beginning, the flow is defined on $t \in [0, 1]$. Later on, the index is shifted to $\{0, …, T\}$.
- Around equation (2): it is unclear whether $s \in [0, T]$ or $s \in [1, T]$.
- Equation (3) and (4): How could I sample $j \sim \mathcal U(s, j)$?
- Algorithm 1, line 14: the EMA update is incorrect.
- Algorithm 2, line 6: Do we miss a $\Delta k$ term?
- Proposition 1 only holds under some assumptions of the noise distribution $\pi$. These assumptions have not been made explicit.

**Questions:**

- How sensitive is the method to the choice of the split points between sub-paths?

- Does the performance degrade significantly if the initial diffusion model is not well-trained?

- Can FUM be trained entirely from scratch without relying on a pre-trained ODE sampler?

- What is the computational overhead compared to MeanFlow or Shortcut models during training?

- Could the velocity uniqueness concept be extended to text-to-image or multimodal setups?

---

> ### Author Response · Authors · 2025-11-21
>
> ### **Response to W1: Limited novelty-the proposed approach mainly combines existing components (pairwise matching + consistency regularization) without introducing a fundamentally new principle.**
>
> **Novelty Contribution.** Our method is not a simple combination of pairwise matching and consistency regularization; instead, it introduces a new principle for one-step generative modeling: constructing a unique instantaneous velocity field by design.
> Specifically, we propose a novel Flow Uniqueness Model (FUM) that exploits strictly one-to-one pairs induced by the deterministic ODE sampler.
> These pairs are generated via forward diffusion rather than by a pre-trained diffusion model (DM), avoiding extra computational cost and ensuring that the paired samples lie on the ground-truth path, free from truncation errors typical of model-generated pairs.
> Under this bijective correspondence, the velocity associated with each time instant is uniquely defined, representing a single specific flow rather than a mixture of multiple flows (cf. Figure 2 in MeanFlow [1]).
>
> **Why this differs from standard pairwise matching.**
> Pairwise matching is indeed central to flow matching and recent one-step methods such as MeanFlow [1] and Shortcut models [2].
> However, they typically form noise-image pairs randomly (or via model-generated trajectories), so the learned velocity at a given time can correspond to multiple possible flows.
> In contrast, FUM enforces a deterministic one-to-one correspondence, making the velocity field single-flow consistent at each time instant.
> This uniqueness property is the key ingredient that enables faithful one-step modeling.
>
> **Why our flow consistency strategy is different.**
> We further introduce flow consistency strategies to extend local velocity uniqueness to the entire sampling path.
> Concretely, we enforce velocity consistency between two sub-paths by exploiting the additivity of integrals.
> This differs from classical consistency models [3], which learn a direct mapping from any point on the trajectory to its endpoint.
> In contrast, our velocity consistency defines a distinct form of consistency tailored to one-step, flow-based modeling, and enables a controllable trade-off between image quality and sampling NFEs-flexibility that standard consistency models do not provide.
>
> Overall, our framework contributes a fundamentally new perspective on how velocities are defined and learned for one-step generative modeling: unique velocities induced by deterministic bijective pairing, then globalized by sub-path flow consistency.
> Extensive experiments validate the effectiveness of this principle and suggest it can inspire future one-step methods with improved performance.
>
> [1] Mean Flows for One-step Generative Modeling. NeurIPS 2025.
>
> [2] One Step Diffusion via Shortcut Models. ICLR 2025.
>
> [3] Consistency Models. ICML 2023.

---

> ### Author Response · Authors · 2025-11-21
>
> ### **Response to W2&Q4: A fully end-to-end training scheme and its training computational overhead.**
> We agree that FUM is not primarily designed as a standalone, from-scratch end-to-end training scheme.
> Instead, it is intentionally built as a lightweight, plug-in module on top of a pre-trained DM, which is a common and practical setting in modern generative modeling.
>
> We employ a pre-trained DM for two reasons:
> (1) **Strict one-to-one pair construction.**
> We leverage the deterministic nature of the ODE sampler to induce bijective image pairs.
> Concretely, these paired samples are obtained via forward diffusion, without any additional training of the backbone.
> (2) **Computational efficiency.**
> FUM is trained only as a lightweight velocity head on top of the already-trained backbone, making its added optimization cost modest compared with end-to-end one-step methods.
> To quantify this advantage, we compare training budgets against current end-to-end one-step baselines (including the cost of the base EDM pretraining for FUM, which is a conservative accounting):
> |Models|FID|NFE|Training Iterations|Batch size|
> |---|---|---|---|---|
> |IMM [1]|3.20|1|400k|4096|
> |Meanflow [2]|2.92|1|800k|1024|
> |Ours|2.01|1|400k (EDM)+40k(FUM)|512|
>
> Despite counting the backbone pretraining, FUM requires only 40k additional iterations with a small batch size to reach substantially better one-step performance (FID 2.01), implying a significantly lower additional compute burden than end-to-end one-step baselines.
> Thus, the dependence on a pre-trained DM is a deliberate design choice: we reuse a strong diffusion backbone and add a small, efficient FUM stage, rather than retraining an entire one-step model from scratch.
>
> Moreover, to verify that FUM can also be trained end-to-end, we conducted preliminary from-scratch experiments on CIFAR-10:
> |Models|FID|NFE|Training Iterations|Batch size|
> |---|---|---|---|---|
> |IMM [1]|3.20|1|400k|4096|
> |Meanflow [2]|2.92|1|800k|1024|
> |Ours (employing pretrained EDM)|2.01|1|400k (EDM)+40k(FUM)|512|
> |Ours (training from scratch)|2.69|1|750k|512|
> |Ours (training from scratch)|2.64|1|800k|512|
> |Ours (training from scratch)|2.53|1|600k|1024|
> |Ours (training from scratch)|2.41|1|800k|1024|
>
> FUM consistently achieves competitive one-step performance when trained from scratch, which further supports its generality.
> We believe this setting can be improved with more specialized end-to-end training strategies, which we leave for future work.
>
> [1] Inductive Moment Matching. ICML 2025.
>
> [2] Mean Flows for One-step Generative Modeling. NeurIPS 2025.
>
> [3] One Step Diffusion via Shortcut Models. ICLR 2025.
>
> ### **Response to W3: 'Strong Uniqueness'.**
> We thank the reviewer for pointing out the ambiguity of this phrase.
> Our use of 'strong uniqueness' is intended as a qualitative, structural property rather than a scalar quantity measured by a specific numerical metric.
>
> In standard flow-matching frameworks, noise-image pairs $(\epsilon, x_{0})$ are constructed randomly.
> As a result, an intermediate point $x_{t}$ obtained by time interpolation between $(\epsilon, x_{0})$ can be produced by many different pairings, so the velocity at time $t$ may correspond to multiple underlying flows (see Figure 2 of MeanFlow [1]).
> In other words, the velocity field at a given time is not uniquely tied to a single flow.
> In contrast, we construct strict one-to-one correspondence pairs $(x_{s}, x_{0})$.
> Therefore, any interpolated point $x_{t}$ between $(x_{s}, x_{0})$ is associated with a single specific pair, inducing a single specific velocity direction at each time.
> When combined with our flow-consistency strategy, this one-to-one uniqueness is further extended from the local segment $(x_{s}, x_{0})$ to the entire path $(\epsilon, x_{0})$ by exploiting the additivity of integrals.
> Consequently, the velocity induced by our constructed pairs is unambiguous: each pair defines exactly one flow and one velocity field along time.
>
> By 'strong uniqueness' we precisely mean this one-to-one, unambiguous association between a correspondence pair and its induced velocity field, in contrast to the potentially many-to-one association in standard flow matching.
> Since this is a structural property of the pairing construction rather than a scalar measure, it is not directly quantified by a numerical metric.
>
> [1] Mean Flows for One-step Generative Modeling. NeurIPS 2025.
>
> ### **Response to W4: Typos and inconsistencies.**
> We have revised all the typos and inconsistencies.

---

> ### Author Response · Authors · 2025-11-21
>
> ### **Response to Q1: Split points sensitive.**
> To test the sensitivity to the split point, we conduct additional ablations on ImageNet-256 using the pre-trained DiT-XL/2 [1].
> We discretize the sampling horizon into 50 EDM scheduler steps [2].
> Empirically, we find that $t=15$ is the largest split point for which the sampler can still denoise $x\_{t}$ back to $\tilde{x}\_{0}$ that remains highly semantically consistent with $x\_{0}$. We therefore test $t=\{7,9,11,13,15,17\} $ and report one-step (1 NFE) FID results below:
>
> |$t$|7|9|11|13|15|17|
> |---|---|---|---|---|---|---|
> |FUM+Shortcut consistency|10.64|6.70|5.19|3.87|3.23|80.36|
>
> As shown, the best performance is achieved at $t=15$.
> Within the feasible range $t\le 15$, performance degrades gradually as $t$ becomes smaller.
> We attribute this trend to the fact that maintaining one-to-one uniqueness over a longer early segment yields smoother and more reliable velocity consistency over the full path.
> When $t=17$, performance drops sharply because the base sampler can no longer reliably denoise $x\_{t}$ to an output that is semantically similar to $x\_{0}$, violating the prerequisite for constructing valid one-to-one pairs.
>
> In summary, FUM is not overly sensitive to the split point within the feasible range and benefits from choosing the largest split time at which the base sampler still denoises reliably (here, $t=15$).
>
> [1] Scalable Diffusion Models with Transformers. ICCV 2023.
>
> [2] Elucidating the Design Space of Diffusion-Based Generative Models. NeurIPS 2022.
>
>
> ### **Response to Q2: Initialize with a not well-trained diffusion model.**
> To assess whether FUM degrades significantly when the initial DM is not well trained, we consider an extreme setting in which no reliable pre-trained DM is available.
> In this case, we train FUM from scratch with randomly initialized parameters, i.e., without leveraging any pre-trained backbone.
> We conduct preliminary experiments on CIFAR-10, with results reported below:
> |Models|FID|NFE|Training Iterations|Batch size|
> |---|---|---|---|---|
> |IMM [1]|3.20|1|400k|4096|
> |Meanflow [2]|2.92|1|800k|1024|
> |Ours (employing pretrained EDM)|2.01|1|400k (EDM)+40k(FUM)|512|
> |Ours (training from scratch)|2.69|1|750k|512|
> |Ours (training from scratch)|2.64|1|800k|512|
> |Ours (training from scratch)|2.53|1|600k|1024|
> |Ours (training from scratch)|2.41|1|800k|1024|
>
> As expected, training from scratch yields weaker performance than using a well-trained EDM backbone (FID 2.01 -> best scratch FID 2.41).
> However, even in this extreme scenario, FUM remains competitive and still outperforms recent one-step baselines, including IMM [1] and MeanFlow [2].
> This suggests that while a stronger backbone improves absolute performance, FUM does not collapse under imperfect initialization and retains its advantage.
> We believe from-scratch performance can be further improved with stronger end-to-end training techniques (e.g., improved schedules, regularization, or larger-scale tuning), which we leave for future work.
>
> [1] Inductive Moment Matching. ICML 2025.
>
> [2] Mean Flows for One-step Generative Modeling. NeurIPS 2025.
>
>
> ### **Response to Q5: Text-to-image performance.**
> To verify this, we conduct preliminary experiments on MS COCO 2017 based on a pre-trained SD [1], with the detailed results shown below.
> |Models|FID-5k|CLIP|NFE|
> |---|---|---|---|
> |SD [1]|20.1|0.318|25|
> |InstaFlow [2]|23.4|0.304|1|
> |Ours|22.5|0.311|1|
>
> These results show that our method achieves competitive one-step text-to-image performance, comparable to a representative one-step baseline (InstaFlow [2]), which supports the practicality of extending velocity uniqueness to multimodal generation.
>
> [1] High-Resolution Image Synthesis with Latent Diffusion Models. CVPR 2022.
>
> [2] InstaFlow: One Step is Enough for High-Quality Diffusion-Based Text-to-Image Generation. ICLR 2024.

---

### Author Response · Authors · 2025-11-25

Dear Reviewers and ACs,

As the discussion period comes to an end, we would like to express our sincere gratitude to all reviewers for their constructive feedback and to the ACs for their efforts in facilitating the discussion.

We especially appreciate the time and care the reviewers devoted to the first-round evaluation.

In response, we have carefully revised the manuscript and updated the PDF; all changes are highlighted in brown.
Specifically, we (i) add further analysis to clarify our motivation for proposing FUM, (ii) provide a detailed discussion of how to divide the full flow path, and (iii) strengthen the theoretical justification to better support the effectiveness of our framework.
Moreover, we include additional experiments to verify the robustness of FUM, including text-to-image evaluations, split-time sensitivity ablations, and image-diversity assessments; all results are reported in the Appendix.

In closing, we again thank the reviewers and ACs for their valuable feedback, and we respectfully ask you to reconsider your evaluation of our paper.

Warm regards,

Authors

---

### Meta-Review · Area_Chair_gh5T · 2026-01-06

**Summary:**

This paper proposes an interesting flow matching framework that involves two subpaths. The first subpath is based on a deterministic noise-image pair generated from the forward process, unlike other one-step methods. The velocity of the second subpath is matched to that of the first subpath with consistency model.

While reviewers appreciate the importance of the problem and modest improvement over other models, the main concerns are the lack of technical novelty, in particular, without a sound theory to motivate such an approach with two sub-paths. The authors have addressed many other concerns in the rebuttal. The authors are encouraged to better motivate the method, improve the theory, and resubmit the paper to a future venue.

**Reviewer Concerns:**

Concerns that were addressed by the rebuttal:
- The method relies on pretrained models (Reviewer aCia). The authors have included additional experiments of training from scratch. The effectiveness is demonstrated in both settings of fine-tuning and training from scratch.
- Missing computational analysis of overhead over MeanFlow or Shortcut models. (Reviewer aCia, a4Z5) The authors have reported such overhead in the revised version.
- Applicability to text-to-image models (Reviewer acia, a4Z5). The authors have included new experiments on text-to-image models.
- Relying on strictly one-to-one pairs may hinder the diversity of image generation (reviewer co9r). The authors have further included diversity metrics in the rebuttal. However, no theoretical guarantee is provided.
- Sensitivity of the choice of R (reviewer VjJE). The authors have included an ablation study on R.

Outstanding concerns:
- limited novelty (Reviewer aCia, co9r). Multiple reviewers raised the concern that the method here is technically sound, but it seems incremental by choosing deterministic noise-image pairs and enforcing consistency between the first sub-path and the second sub-path.
- Lacking a theory on why the proposed method with two subpaths is better than single path method. The motivation of temporal split is not clear. (reviewer co9r, VjJE)

**Reviewer Scores:**

Reviewer aCia is likely to increase the score from 2 to 4.

Reviewer a4Z5 is likely to keep the score of 6.

Reviewer co9r is likely to keep the score of 4.

Reviewer VjJE is likely to keep the score of 4.

---

### Decision · Program_Chairs · 2026-01-26

Reject